# Efficient Algorithms for Smooth Minimax Optimization

**Kiran Koshy Thekumprampil**
University of Illinois at Urbana-Champaign
thekump2@illinois.edu

**Prateek Jain**
Microsoft Research, India
prajain@microsoft.com

**Praneeth Netrapalli**
Microsoft Research, India
praneeth@microsoft.com

**Sewoong Oh**
University of Washington, Seattle
sewoong@cs.washington.edu

## Abstract

This paper studies first order methods for solving smooth minimax optimization problems $\min_x \max_y g(x, y)$ where $g(\cdot, \cdot)$ is smooth and $g(x, \cdot)$ is concave for each $x$. In terms of $g(\cdot, y)$, we consider two settings – strongly convex and nonconvex – and improve upon the best known rates in both. For strongly-convex $g(\cdot, y)$, $\forall y$, we propose a new direct optimal algorithm combining Mirror-Prox and Nesterov's AGD, and show that it can find global optimum in $\widetilde{O}\left(1/k^2\right)$ iterations, improving over current state-of-the-art rate of $O(1/k)$. We use this result along with an inexact proximal point method to provide $\widetilde{O}\left(1/k^{1/3}\right)$ rate for finding stationary points in the nonconvex setting where $g(\cdot, y)$ can be nonconvex. This improves over current best-known rate of $O(1/k^{1/5})$. Finally, we instantiate our result for finite nonconvex minimax problems, i.e., $\min_x \max_{1 \le i \le m} f_i(x)$, with nonconvex $f_i(\cdot)$, to obtain convergence rate of $O(m^{1/3}\sqrt{\log m}/k^{1/3})$.

## 1 Introduction

In this paper we study smooth minimax problems of the form:

$$\min_{x \in \mathcal{X}} \max_{y \in \mathcal{Y}} \ g(x, y) \ , \ \ g : \mathcal{X} \times \mathcal{Y} \to \mathbb{R}, \ g \text{ is smooth i.e., gradient Lipschitz.} \quad (1)$$

The problem has applications in several domains such as machine learning [15, 29], optimization [5], statistics [3], mathematics [23], and game theory [31]. Given the importance of these problems, there is an extensive body of work that studies various algorithms and their convergence properties. The vast majority of existing results for this problem focus on the convex-concave setting, where $g(\cdot, y)$ is convex for every $y$ and $g(x, \cdot)$ is concave for every $x$. The best known convergence rate in this setting is $O(1/k)$ for the primal-dual gap, achieved for example by Mirror-Prox [34]. This rate is also known to be optimal for the class of smooth convex-concave problems [41]. A natural question is whether we can achieve a faster convergence if we have strong convexity (as opposed to just convexity) of $g(\cdot, y)$. We answer this in the affirmative, by introducing an algorithm that achieves a convergence rate of $\widetilde{O}\left(1/k^2\right)$ for the general smooth, strongly-convex–concave minimax problem. The algorithm we propose is a novel combination of Mirror-Prox and Nesterov's accelerated gradient descent. This matches the known lower bound of $\Omega(1/k^2)$ from [41], closing the gap up to a poly-logarithmic factor. There also exists a conceptually simple smoothing technique based indirect algorithm, which prefixes the tolerance of $\varepsilon$. However, our goal is to find a direct algorithm which does not prefix the tolerance. Other known methods that obtain a rate of $O(1/k^2)$ in this context are for very special cases, where $x$ and $y$ are connected through a bi-linear term or $g(x, \cdot)$ is linear in $y$ [35, 20, 14, 8, 49, 16, 48].

| Setting | Optimality notion | Previous state-of-the-art | Our results | Smoothing schemes | Lower bound |
|---|---|---|---|---|---|
| Convex | Primal-dual gap | $O\left(k^{-1}\right)$ [34] | - | - | $\Omega(k^{-1})$ [41] |
| Strongly convex | Primal-dual gap | $O\left(k^{-1}\right)$ [34] | $\widetilde{O}\left(k^{-2}\right)$ | $\widetilde{O}\left(k^{-2}\right)$ | $\Omega(k^{-2})$ [41] |
| Nonconvex | Approx. stat. point | $O\left(k^{-1/5}\right)$ [18] | $\widetilde{O}\left(k^{-1/3}\right)$ | $\widetilde{O}\left(k^{-1/3}\right)$ [26] | - |

Table 1: Comparison of our results with previous state-of-the-art. We assume that $g(\cdot,\cdot)$ is smooth (i.e., has Lipschitz gradients) and $g(x,\cdot)$ is concave $\forall x \in \mathcal{X}$. Convexity, strong convexity and nonconvexity in the first column refers to $g(\cdot,y)$ for fixed $y$. Smoothing schemes are indirect methods using the smoothing technique [36].

While most theoretical results focus on the convex-concave setting, several real world problems fall outside this class. A slightly larger class, which captures several more applications, is the class of smooth nonconvex–concave minimax problems, where $g(x,\cdot)$ is *concave* for every $x$ but $g(\cdot,y)$ can be nonconvex. For example, finite minimax problems, i.e., $\min_x \max_{i=1}^m f_i(x) = \min_x \max_{0 \preceq y \preceq 1, \sum_{i=1}^m y_i=1} \sum_i y_i \cdot f_i(x) := g(x,y)$ belong to this class, and so do smooth non-convex constrained optimization problems [25]. In addition, several machine learning problems with non-decomposable loss functions [22] also belong to this class.

In this general nonconvex concave setting however, we cannot hope to find global optimum efficiently as even the special case of nonconvex optimization is NP-hard. Similar to nonconvex optimization, we might hope to find an approximate stationary point [37].

Our second contribution is a new algorithm and a faster rate for the *general smooth nonconvex–concave* minimax problem. Our algorithm is an inexact proximal point method for the nonconvex function $f(x) := \max_{y \in \mathcal{Y}} g(x,y)$. The key insight is that the proximal point problem in each iteration results in a strongly-convex concave minimax problem, for which we use our improved algorithm to obtain the overall computation/iteration complexity of $\widetilde{O}\left(1/k^{1/3}\right)$ thus improving over the previous best known rate of $O(1/k^{1/5})$ [18][1]. More recently, independent to our work, a smoothing based algorithm has also been proposed to achieve the same $O\left(k^{-1/3}\right)$ rate [26].

Finally, we specialize our result to finite minimax problems, i.e., $\min_x \max_{1 \le i \le m} f_i(x)$ where $f_i(x)$ can be nonconvex function but each $f_i$ is a smooth function; nonconvex constrained optimization problems can be reduced to such finite minimax problems. For these, we obtain a rate of $\widetilde{O}\left(m^{1/3}\sqrt{\log m}/k^{1/3}\right)$ total gradient computations which improves upon the state-of-the-art rate $O(m^{1/4}/k^{1/4})$ [11] in this setting as well.

**Summary of contributions**: See also Table 1.
1. Optimal $\widetilde{O}\left(1/k^2\right)$ convergence rate for smooth, strongly-convex – concave problems, improving upon the previous best known rate of $O\left(1/k\right)$ for a direct algorithm and,
2. $\widetilde{O}\left(1/k^{1/3}\right)$ convergence rate for smooth, nonconvex – concave problems, improving upon the previous best known rate of $O\left(1/k^{1/5}\right)$.

**Related works**: For strongly-convex-concave minimax problems with special structures, several algorithms have been proposed. In an increasing order of generality, [14, 49, 50] study optimizing a strongly convex function with linear constraints, which can be posed as a special case of minimax optimization, [35, 8] study a minimax problem where $x$ and $y$ are connected only through a bi-linear term $y^T A x$, and [16, 20] study a case where $g(x,\cdot)$ is linear in $y$. In all these cases, it is shown that $O(1/k^2)$ convergence rate is achievable if $g(\cdot,y)$ is strongly-convex $\forall\, y$. Recently, [12] showed linear convergence of gradient descent ascent for strongly-convex–concave problem with bilinear coupling when $A$ has full row rank. However, it has remained an open question if the fast rate of $O(1/k^2)$ can be achieved for general strongly-convex-concave minimax problems. See [32, 9, 7, 17, 51, 1]

for detailed surveys on the results for the convex-concave minimax problems. For examples and application of saddle point problems refer [36, 19, 20, 7, 43].

For nonconvex-concave minimax problems, [42] considers both deterministic and stochastic settings, and proposes inexact proximal point methods for solving smooth nonconvex–concave problems. In the deterministic setting, their result guarantees an error of $O(1/k^{1/6})$. We note that there have also been other notions of stationarity proposed in literature for nonconvex-concave minimax problems [28, 40]. These notions however are weaker than the one considered in this paper, in the sense that, our notion of stationarity implies these other notions (without loss in parameters). For one such weaker notion, [40] proposes an algorithm with a convergence rate of $O\left(k^{-1/3.5}\right)$. Since the notion they consider is weaker, it does not imply the same convergence rate in our setting.

We would also like to highlight the works [6, 13, 33, 46, 34, 47, 10] designing efficient algorithms for solving monotone variational inequalities which generalizes the convex-concave minimax problems.

**Notations**: $\mathbb{R}$ is the real line and for any natural number $p$, $\mathbb{R}^p$ is the real vector space of dimension $p$. $\|\cdot\|$ is a norm on some metric space which would be evident from the context. For a convex set $\mathcal{X} \subseteq \mathbb{R}^p$ and $x \in \mathbb{R}^p$, $\mathcal{P}_{\mathcal{X}}(x) = \arg\min_{x' \in \mathcal{X}} \|x - x'\|$ is the projection of $x$ on to $\mathcal{X}$. For a differentiable function $g(x, y)$, $\nabla_x g(x, y)$ is its gradient with respect to $x$ at $(x, y)$. We use the standard big-O notations. For functions $T, S : \mathbb{R} \to \mathbb{R}$ such that $0 < \liminf_{x \to \infty} T(x)$, $0 < \liminf_{x \to \infty} S(x)$, (a) $T(x) = O(S(x))$ means $\limsup_{x \to \infty} T(x)/S(x) < \infty$; (b) $T(x) = \Theta(S(x))$ means $T(x) = O(S(x))$ and $S(x) = O(T(x))$; and (c) $T(x) = \widetilde{O}(S(x))$ means that $T(x) = O(S(x)R(x))$ for some poly-logarithmic function $R : \mathbb{R} \to \mathbb{R}$.

**Paper organization**: In Section 2, we present preliminaries and all relevant background. In Section 3, we present our results for strongly-convex–concave setting and in section 4, results for nonconvex–concave setting. In Section 5, we present empirical evaluation of our algorithm for nonconvex-concave setting and compare it to a state-of-the-art algorithm. We conclude in Section 6. Several technical details are presented in the appendix.

## 2 Preliminaries and background material

In this section, we will present some preliminaries, describing the setup and reviewing some background material that will be useful in the sequel.

### 2.1 Minimax problems

We are interested in the minimax problems of the form (1) where $g(x, y)$ is a smooth function.

**Definition 1.** *A function $g(x, y)$ is said to be L-smooth if:*

$$\max\left\{\|\nabla_x g(x, y) - \nabla_x g(x', y')\|, \|\nabla_y g(x, y) - \nabla_y g(x', y')\|\right\} \leq L\left(\|x - x'\| + \|y - y'\|\right).$$

Throughout, we assume that $g(x, .)$ is *concave* for every $x \in \mathcal{X}$. For $g(\cdot, y)$ behavior in terms of $x$, there are broadly two settings:

### 2.1.1 Convex-concave setting

In this setting, $g(\cdot, y)$ is convex $\forall\, y \in \mathcal{Y}$. Given any $g$ and $\forall(\widehat{x}, \widehat{y})$, the following holds trivially:

$$\min_{x \in \mathcal{X}} g(x, \widehat{y}) \leq g(\widehat{x}, \widehat{y}) \leq \max_{y \in \mathcal{Y}} g(\widehat{x}, y),$$

which then implies that $\max_{y \in \mathcal{Y}} \min_{x \in \mathcal{X}} g(x, y) \leq \min_{x \in \mathcal{X}} \max_{y \in \mathcal{Y}} g(x, y)$. The celebrated minimax theorem for the convex-concave setting [44] says that if $\mathcal{Y}$ is a compact set then the above inequality is in fact an equality, i.e., $\max_{y \in \mathcal{Y}} \min_{x \in \mathcal{X}} g(x, y) = \min_{x \in \mathcal{X}} \max_{y \in \mathcal{Y}} g(x, y)$. Furthermore, any point $(x^*, y^*)$ is an optimal solution to (1) if and only if:

$$\min_{x \in \mathcal{X}} g(x, y^*) = g(x^*, y^*) = \max_{y \in \mathcal{Y}} g(x^*, y). \tag{2}$$

Hence, our goal is to find $\varepsilon$-primal-dual pair $(\widehat{x}, \widehat{y})$ with small primal-dual gap: $\max_{y \in \mathcal{Y}} g(\widehat{x}, y) - \min_{x \in \mathcal{X}} g(x, \widehat{y})$.

**Definition 2.** *For a convex-concave function $g : \mathcal{X} \times \mathcal{Y} \to \mathbb{R}$, $(\hat{x}, \hat{y})$ is an $\varepsilon$-primal-dual-pair of $g$ if the primal-dual gap is less than $\varepsilon$: $\max_{y \in \mathcal{Y}} g(\widehat{x}, y) - \min_{x \in \mathcal{X}} g(x, \widehat{y}) \leq \varepsilon$.*

### 2.1.2 Nonconvex-concave setting

In this setting the function $g(\cdot, y)$ need not be convex. One cannot hope to solve such problems in general, since the special case of nonconvex optimization is already NP-hard [39]. Furthermore, the minimax theorem no longer holds, i.e., $\max_{y \in \mathcal{Y}} \min_{x \in \mathcal{X}} g(x, y)$ can be strictly smaller than $\min_{x \in \mathcal{X}} \max_{y \in \mathcal{Y}} g(x, y)$, and therefore the order of $min$ and $max$ might be important for a given application i.e., we might be interested only in minimax but not maximin (or vice versa). So, the primal-dual gap may not be a meaningful quantity to measure convergence. In this paper we will focus on the minimax problem: $\min_{x \in \mathcal{X}} \max_{y \in \mathcal{Y}} g(x, y)$. One approach, inspired by nonconvex optimization, to measure convergence is to consider the function $f(x) = \max_{y \in \mathcal{Y}} g(x, y)$ and consider the convergence rate to approximate first order stationary points (i.e., $\nabla f(x)$ is small)[42, 18]. But as $f(x)$ could be non-smooth, $\nabla f(x)$ might not even be defined. It turns out that whenever $g(x, y)$ is smooth, $f(x)$ is weakly convex (Definition 4) for which first order stationarity notions are well-studied and are discussed below.

**Approximate first-order stationary point for weakly convex functions**: We first need to generalize the notion of gradient for a non-smooth function.

**Definition 3.** *The Fréchet sub-differential of a function $f(\cdot)$ at $x$ is defined as the set, $\partial f(x) = \{u \mid \liminf_{x' \to x} f(x') - f(x) - \langle u, x' - x \rangle / \|x' - x\| \geq 0\}$.*

In order to define approximate stationary points, we also need the notion of weakly convex function and Moreau envelope.

**Definition 4.** *A function $f : \mathcal{X} \to \mathbb{R} \cup \{\infty\}$ is $L$-weakly convex if,*

$$f(x) + \langle u_x, x' - x \rangle - \frac{L}{2} \|x' - x\|^2 \ \leq \ f(x'), \tag{3}$$

*for all Fréchet subgradients $u_x \in \partial f(x)$, for all $x, x' \in \mathcal{X}$.*

**Definition 5.** *For a proper lower semi-continuous (l.s.c.) function $f : \mathcal{X} \to \mathbb{R} \cup \{\infty\}$ and $\lambda > 0$ ($\mathcal{X} \subseteq \mathbb{R}^p$), the Moreau envelope function is given by*

$$f_\lambda(x) \ = \ \min_{x' \in \mathcal{X}} f(x') + \frac{1}{2\lambda} \|x - x'\|^2 . \tag{4}$$

Lemma 4 (in Appendix B.2) provides some useful properties of the Moreau envelope for weakly convex functions. Now, first order stationary point of a non-smooth nonconvex function is well-defined, i.e., $x^*$ is a *first order stationary point (FOSP)* of a function $f(x)$ if, $0 \in \partial f(x^*)$ (see Definition 3). However, unlike smooth functions, it is nontrivial to define an *approximate* FOSP. For example, if we define an $\varepsilon$-FOSP as the point $x$ with $\min_{u \in \partial f(x)} \|u\| \leq \varepsilon$, there may never exist such a point for sufficiently small $\varepsilon$, unless $x$ is exactly a FOSP. In contrast, by using above properties of the Moreau envelope of a weakly convex function, it's approximate FOSP can be defined as [11]:

**Definition 6.** *Given an $L$-weakly convex function $f$, we say that $x^*$ is an $\varepsilon$-first order stationary point ($\varepsilon$-FOSP) if, $\|\nabla f_{\frac{1}{2L}}(x^*)\| \leq \varepsilon$, where $f_{\frac{1}{2L}}$ is the Moreau envelope with parameter $1/2L$.*

Using Lemma 4, we can show that for any $\varepsilon$-FOSP $x^*$, there exists $\hat{x}$ such that $\|\hat{x} - x^*\| \leq \varepsilon/2L$ and $\min_{u \in \partial f(\hat{x})} \|u\| \leq \varepsilon$. In other words, an $\varepsilon$-FOSP is $O(\varepsilon)$ close to a point $\hat{x}$ which has a subgradient smaller than $\varepsilon$. We note that other notions of FOSP have also been proposed recently such as in [40]. However, it can be shown that an $\varepsilon$-FOSP according to the above definition is also an $\epsilon$-FOSP with [40]'s definition as well, but the reverse is not necessarily true.

## 2.2 Mirror-Prox

Mirror-Prox [34] is a popular algorithm proposed for solving convex-concave minimax problems (1). It achieves a convergence rate of $O(1/k)$ for the primal dual gap. The original Mirror-Prox paper [34] motivates the algorithm through a *conceptual* Mirror-Prox (CMP) method, which brings out the main idea behind its convergence rate of $O(1/k)$. CMP for Euclidean norm (after ignoring projections to $\mathcal{X}$ and $\mathcal{Y}$) does the following update:

$$(x_{k+1}, y_{k+1}) = (x_k, y_k) + \frac{1}{\beta} (-\nabla_x g(x_{k+1}, y_{k+1}), \nabla_y g(x_{k+1}, y_{k+1})). \tag{5}$$

The main difference between CMP and standard gradient descent ascent (GDA) is that in the $k^{\text{th}}$ step, while GDA uses gradients at $(x_k, y_k)$, CMP uses gradients at $(x_{k+1}, y_{k+1})$. The key observation of [34] is that if $g(\cdot, \cdot)$ is smooth, it can be implemented efficiently. CMP is analyzed as follows:

**Implementability of CMP**: Let $(x_k^{(0)}, y_k^{(0)}) = (x_k, y_k)$. For $\beta < \frac{1}{L}$, the iteration

$$\left(x_k^{(i+1)}, y_k^{(i+1)}\right) = (x_k, y_k) + \frac{1}{\beta}\left(-\nabla_x g\left(x_k^{(i)}, y_k^{(i)}\right), \nabla_y g\left(x_k^{(i)}, y_k^{(i)}\right)\right). \tag{6}$$

can be shown to be $\frac{1}{\sqrt{2}}$-contraction (when $g(\cdot, \cdot)$ is smooth) and that its fixed point is $(x_{k+1}, y_{k+1})$. So, in $\log \frac{1}{\epsilon}$ iterations of (6), we can obtain an accurate version of the update required by CMP. In fact, [34] showed that just *two* iterations of (6) suffice [30].

**Convergence rate of CMP**: Using CMP update with simple manipulations leads to the following:

$$g(x_{k+1}, y) - g(x, y_{k+1}) \leq \frac{\beta}{2}\left(\|x - x_k\|^2 - \|x - x_{k+1}\|^2 + \|y - y_k\|^2 - \|y - y_{k+1}\|^2\right), \forall x \in \mathcal{X}, \ y \in \mathcal{Y}.$$

$O(1/k)$ convergence rate follows easily using the above result.

Finally, our method and analysis also requires Nesterov's accelerated gradient descent method (see Algorithm 3 in Appendix A) and it's per-step analysis by [2] (Lemma 2 in Appendix A).

## 3   Strongly-convex concave saddle point problem

We first study the minimax problem of the form:

$$\min_{x \in \mathcal{X}}\left[\ f(x) = \max_{y \in \mathcal{Y}} g(x, y)\ \right], \tag{P1}$$

where $g(x, \cdot)$ is concave, $g(\cdot, y)$ is $\sigma$-*strongly-convex*, $g(\cdot, \cdot)$ is $L$-smooth, i.e., $0 < \sigma \leq L$. $\mathcal{X} = \mathbb{R}^p$ and $\mathcal{Y} \subset \mathbb{R}^q$ is a convex compact sub-set of $\mathbb{R}^q$ and let the function $f$ take a minimum value $f^*(> -\infty)$. Let $D_{\mathcal{Y}} = \max_{y, y' \in \mathcal{Y}} \|y - y'\|$ be the diameter of $\mathcal{Y}$.

Our objective here is to find an $\epsilon$-primal-dual pair $(\widehat{x}, \widehat{y})$ (see Definition 2). Now the fact that $f(\hat{x}) - f^* \leq \max_{y \in \mathcal{Y}} g(\widehat{x}, y) - \min_{x \in \mathcal{X}} g(x, \widehat{y})$ implies that if $(\hat{x}, \hat{y})$ is an $\varepsilon$-primal-dual-pair, then $\hat{x}$ is also an $\varepsilon$-approximate minima of $f$. Furthermore, by Sion's minimax theorem [24], strong-convexity–concavity of $g(\cdot, \cdot)$ ensures that: $\min_x[f(x) := \max_y g(x, y)] = \max_y[h(y) := \min_x g(x, y)]$. Hence, one approach to efficiently solving the problem is by optimizing the dual problem $\max_y h(y)$. By Lemma 6 (in Appendix B.6), $h(y)$ is an $(L + \frac{L^2}{\sigma})$-smooth function. So we can use AGD to ensure that $h(y_k) - h(y^*) = O(1/k^2)$. Now, each step of AGD requires computing $\arg\min_x g(x, y_k)$ which can be done efficiently (i.e., logarithmic number of steps) as $g(\cdot, y_k)$ is strongly-convex and smooth. So, the overall first-order oracle complexity is $h(y_k) - h(y^*) = \widetilde{O}\left(1/k^2\right)$.

So does this simple approach give us our desired result? Unfortunately that is not the case, as the above bound on the dual function $h$ does not translate to the same error rate for primal function $f$, i.e., the solution need not be $\widetilde{O}\left(1/k^2\right)$-primal-dual pair. E.g., consider $\min_{x \in \mathbb{R}} \max_{y \in [-1,1]}[g(x, y) = xy + x^2/2]$, where $\min_x \max_y g(x, y) = 0$, $f(x) = x^2/2 + |x|$ and $h(y) = -y^2/2$. If $h(y_k) = \Theta(k^{-2})$, then $x_k \in \arg\min_x g(x, y_k) = \Theta(1/k)$ and so $f(x_k)$ is $\Theta(k^{-1})$. This is due to the non-smoothness of $\arg\max_{y \in \mathcal{Y}} g(x, y)$ w.r.t. $x$.

Instead of using AGD, we introduce a new method to solve the dual problem that we refer to as DIAG, which stands for Dual Implicit Accelerated Gradient. DIAG combines ideas from AGD [38] and Nemirovski's original derivation of the Mirror-Prox algorithm [34], and can ensure a fast convergence rate of $\tilde{O}(k^{-2})$ for the primal-dual gap. We note that there also exists a conceptually simpler smoothing technique based indirect algorithm, which prefixes the tolerance of $\varepsilon$ (Appendix D). However, our goal is to find a direct algorithm which does note require prefixing the tolerance at $\varepsilon$. For better exposition, we first present a conceptual version of DIAG (C-DIAG), which is not implementable *exactly*, but brings out the main new ideas in our algorithm. We then present a detailed error analysis for the *inexact* version of this algorithm, which is implementable.

### 3.1   Conceptual version: C-DIAG

Consider the following updates which is a modified version of AGD (see Algorithm 3 in Appendix A):

(a) $w_k = (1 - \tau_k)y_k + \tau_k z_k$

(b) Choose $x_{k+1}, y_{k+1}$ ensuring:
$x_{k+1} \in \arg\min_x g(x, y_{k+1})$, and $y_{k+1} = \mathcal{P}_{\mathcal{Y}}(w_k + \frac{1}{\beta}\nabla_y g(x_{k+1}, w_k))$

(c) $z_{k+1} = \mathcal{P}_{\mathcal{Y}}(z_k + \eta_k \nabla_y g(x_{k+1}, w_k))$

Complete pseudocode for C-DIAG algorithm is presented in Algorithm 4 in Appendix B.4. The main idea of the algorithm is in Step $(b)$ above (i.e., Step 4 of Algorithm 4 in Appendix B.4), where we simultaneously find $x_{k+1}$ and $y_{k+1}$ satisfying the following requirements:

- $x_{k+1}$ is the minimizer of $g(\cdot, y_{k+1})$, and
- $y_{k+1}$ corresponds to an AGD step (see Algorithm 3 in Appendix A) for $g(x_{k+1}, \cdot)$

**Implementability**: The first question is whether it is easy enough to implement such a step? It turns out that it is indeed possible to quickly find points $x_{k+1}$ and $y_{k+1}$ that approximately satisfy the above requirements. The reason is that:

- Since $g(\cdot, y)$ is smooth and strongly convex for every $y \in \mathcal{Y}$, we can find $\epsilon$-approximate minimizer for a given $y$ in $O\left(\log \frac{1}{\epsilon}\right)$ iterations.

- Let $x^*(y) := \operatorname{argmin}_{x \in \mathcal{X}} g(x, y)$. The iteration $y^{i+1} = \mathcal{P}_{\mathcal{Y}}\left(w_k + \frac{1}{\beta}\nabla_y g(x^*(y^i), w_k)\right)$ is a $1/2$-contraction with a unique fixed point satisfying the update step requirements (i.e., Step 4 of Algorithm 4 in Appendix B.4). See Lemma 5 in Appendix B.5 for a proof. This means that only $O\left(\log \frac{1}{\epsilon}\right)$ iterations again suffice to find an update that approximately satisfies the requirements.

**Convergence rate**: Since $y_{k+1}$ and $z_{k+1}$ correspond to an AGD update for $g(x_{k+1}, \cdot)$, we can use the potential function decrease argument for AGD (Lemma 2 in Appendix A) to conclude that $\forall y \in \mathcal{Y}$,

$$(k+1)(k+2)\left(g(x_{k+1}, y) - g(x_{k+1}, y_{k+1})\right) + 2\beta \cdot \|y - z_{k+1}\|^2$$
$$\leq k(k+1)\left(g(x_{k+1}, y) - g(x_{k+1}, y_k)\right) + 2\beta \cdot \|y - z_k\|^2$$
$$\leq k(k+1)\left(g(x_{k+1}, y) - g(x_k, y)\right) + k(k+1)\left(g(x_k, y) - g(x_k, y_k)\right) + 2\beta \cdot \|y - z_k\|^2,$$

where the last step follows from the fact that $x_k = \operatorname{argmin}_x g(x, y_k)$ and so $g(x_k, y_k) \leq g(x_{k+1}, y_k)$. Noting that we can further recursively bound $k(k+1)\left(g(x_k, y) - g(x_k, y_k)\right) + 2\beta \cdot \|y - z_k\|^2$ as above, we obtain

$$(k+1)(k+2)\left(g(x_{k+1}, y) - g(x_{k+1}, y_{k+1})\right) + 2\beta \cdot \|y - z_{k+1}\|^2$$

$$\leq k(k+1)g(x_{k+1}, y) - \sum_{i=1}^{k}(2i) \cdot g(x_i, y) + 2\beta \cdot \|y - z_0\|^2$$

$$\Rightarrow \sum_{i=1}^{k+1}(2i) \cdot g(x_i, y) - (k+1)(k+2)g(x_{k+1}, y_{k+1}) \leq 2\beta \cdot \|y - z_0\|^2.$$

Since $g(x_{k+1}, y_{k+1}) \leq g(x, y_{k+1})$ for every $x \in \mathcal{X}$, we have

$$\sum_{i=1}^{k+1}(2i) \cdot g(x_i, y) - (k+1)(k+2)g(x, y_{k+1}) \leq 2\beta \cdot \|y - z_0\|^2$$

$$\Rightarrow g(\bar{x}_{k+1}, y) - g(x, y_{k+1}) \leq \frac{2\beta \cdot \|y - z_0\|^2}{(k+1)(k+2)},$$

where $\bar{x}_{k+1} := \frac{1}{(k+1)(k+2)}\sum_{i=1}^{k+1}(2i) \cdot x_i$. Since $x$ and $y$ are arbitrary above, this gives a $O\left(1/k^2\right)$ convergence rate for the primal dual gap.

## 3.2 Error analysis

The main issue with Algorithm 4 is that the update step is not exactly implementable. However, as we noted in the previous section, we can quickly find updates that almost satisfy the requirements. Algorithm 1 presents this inexact version. The following theorem states our formal result and a detailed proof is provided in Appendix B.5.

---

**Algorithm 1:** Dual Implicit Accelerated Gradient (DIAG) for strongly-convex–concave programming

---

**Input:** $g$, $L$, $\sigma$, $D_{\mathcal{Y}}$, $x_0$, $y_0$, $K$, $\{\varepsilon_{\text{step}}^{(k)}\}_{k=1}^K$
**Output:** $\bar{x}_K, y_K$

**1** Set $\beta \leftarrow 2\frac{L^2}{\sigma}$, $z_0 \leftarrow y_0$
**2 for** $k = 0, 1, \ldots, K-1$ **do**

**3** $\quad$ $\tau_k \leftarrow \frac{2}{(k+2)}$, $\eta_k \leftarrow \frac{(k+1)}{2\beta}$, $w_k \leftarrow (1-\tau_k)y_k + \tau_k z_k$

**4** $\quad$ $x_{k+1}, y_{k+1} \leftarrow \texttt{Imp-STEP}(g, L, \sigma, x_0, w_k, \beta, \varepsilon_{\text{step}}^{(k+1)})$, ensuring:

$$g(x_{k+1}, y_{k+1}) \leq \min_x g(x, y_{k+1}) + \varepsilon_{\text{step}}^{(k+1)}, \quad y_{k+1} = \mathcal{P}_{\mathcal{Y}}\left(w_k + \frac{1}{\beta}\nabla_y g(x_{k+1}, w_k)\right)$$

**5** $\quad$ $z_{k+1} \leftarrow \mathcal{P}_{\mathcal{Y}}(z_k + \eta_k \nabla_y g(x_{k+1}, w_k))$, $\quad$ $\bar{x}_{k+1} \leftarrow \frac{2}{(k+1)(k+2)}\sum_{i=1}^{k+1} i \cdot x_i$

**6 return** $\bar{x}_K, y_K$

**7** $\texttt{Imp-STEP}(g, L, \sigma, x_0, w, \beta, \varepsilon_{\text{step}})$**:**

**8** $\quad$ Set $\varepsilon_{\text{mp}} \leftarrow \frac{2\sigma}{5L}\sqrt{\frac{2\varepsilon_{\text{step}}}{L}}$, $R \leftarrow \lceil\log_2 \frac{2D_{\mathcal{Y}}}{\varepsilon_{\text{mp}}}\rceil$, $\varepsilon_{\text{agd}} \leftarrow \frac{\sigma\beta^2\varepsilon_{\text{mp}}^2}{32L^2}$, $y_0 \leftarrow w$

**9** $\quad$ **for** $r = 0, 1, \ldots, R$ **do**

**10** $\quad\quad$ Starting at $x_0$ use AGD [38] for strongly-convex $g(\cdot, y_r)$, to compute $\hat{x}_r$ such that:
$$g(\hat{x}_r, y_r) \leq \min_x g(x, y_r) + \varepsilon_{\text{agd}}, \tag{7}$$

**11** $\quad\quad$ $y_{r+1} \leftarrow \mathcal{P}_{\mathcal{Y}}\left(w + \frac{1}{\beta}\nabla_y g(\hat{x}_r, w)\right)$

**12** $\quad$ **return** $\hat{x}_R, y_{R+1}$

---

**Theorem 1** (Convergence rate of DIAG). *Let $g : \mathcal{X} \times \mathcal{Y} \to \mathbb{R}$ be a $L$-smooth, $\sigma$-strongly-convex–concave function on $\mathcal{X} = \mathbb{R}^p$ and a convex compact sub-set $\mathcal{Y} \subset \mathbb{R}^q$ (with diameter $D_{\mathcal{Y}}$). Then, after $K$ iterations, DIAG (Algorithm 1) with a tolerance schedule of $\{\varepsilon_{\text{step}}^{(k)}\}_{k=1}^K$ for its* $\texttt{Imp-STEP}$ *sub-routine, finds $(\bar{x}_K, y_K)$ s.t.:*

$$\max_{\tilde{y} \in \mathcal{Y}} g(\bar{x}_K, \tilde{y}) - \min_{\tilde{x} \in \mathcal{X}} g(\tilde{x}, y_K) \leq \frac{4\frac{L^2}{\sigma}D_{\mathcal{Y}}^2 + \sum_{k=1}^K k(k+1)\varepsilon_{\text{step}}^{(k)}}{K(K+1)}. \tag{8}$$

*In particular, setting $\varepsilon_{\text{step}}^{(k)} = \frac{L^2 D_{\mathcal{Y}}^2}{\sigma k^3(k+1)}$ we have: $\max_{\tilde{y} \in \mathcal{Y}} g(\bar{x}_K, \tilde{y}) - \min_{\tilde{x} \in \mathcal{X}} g(\tilde{x}, y_K) \leq \frac{6\frac{L^2}{\sigma}D_{\mathcal{Y}}^2}{K(K+1)}$.*

*Furthermore, for this setting the total first order oracle complexity is given by: $O(\sqrt{\frac{L}{\sigma}} K \log^2(K))$.*

**Remark 1**: Theorem 1 shows that DIAG needs $\tilde{O}((D_{\mathcal{Y}}L/\sqrt{\sigma\varepsilon}) \cdot (\sqrt{L/\sigma}))$ gradient queries for finding a $\varepsilon$-primal-dual-pair, while current best-known rate is $O(1/\varepsilon)$ achieved by Mirror-Prox. This dependence in $\varepsilon$ and $D_{\mathcal{Y}}$ is optimal, as it is shown in [41, Theorem 10] that $\Omega(D_{\mathcal{Y}}(L - \sigma)/\sqrt{\sigma\varepsilon})$ gradient queries are necessary to achieve $\varepsilon$ error in the primal-dual gap.

**Remark 2**: Unlike standard AGD for $h(y)$, which only updates $y_k$ in the outer-loop, DIAG's outer-step updates both $x_k$ and $y_k$ thus allowing us to better track the primal-dual gap. However, DIAG's dependence on the condition number $L/\sigma$ seems sub-optimal and can perhaps be improved if we do not compute $\texttt{Imp-STEP}$ nearly optimally allowing for inexact updates; we leave further investigation into improved dependence on the condition number for future work.

## 4 Nonconvex concave saddle point problem

We study the nonconvex concave minimax problem (1) where $g(x, \cdot)$ is concave, $g(\cdot, y)$ is nonconvex, and $g(\cdot, \cdot)$ is $L$-smooth, $\mathcal{X} = \mathbb{R}^p$ (such that $\text{Proj}_{\mathcal{X}}(x) = x$) and $\mathcal{Y}$ is a convex compact sub-set of $\mathbb{R}^q$. As mentioned in Section 2, we measure the convergence to an approximate FOSP of this problem (see Definition 6) but it requires weak-convexity of $f(x) := \max_{y \in \mathcal{Y}} g(x, y)$. The following lemma guarantees weak convexity of $f$ given smoothness of $g$.

**Lemma 1.** *Let $g(\cdot, y)$ be continuous and $\mathcal{Y}$ be compact. Then $f(x) = \max_{y \in \mathcal{Y}} g(x, y)$ is $L$-weakly convex, if $g$ is $L$-weakly convex in $x$ (Definition 1), or if $g$ is $L$-smooth in $x$.*

See Appendix B.3 for the proof. The arguments of [18] easily extend to show that applying subgradient method on $f(x)$, [11] gives a convergence rate of $O\left(1/k^{1/5}\right)$. Instead, we exploit the smooth minimax form of $f(\cdot)$ to design a faster converging scheme. The main intuition comes from the proximal viewpoint that gradient descent can be viewed as iteratively forming and optimizing local quadratic upper bounds. As $f$ is weakly convex, adding enough quadratic regularization should ensure that the resulting sequence of problems are all strongly-convex–concave. We then exploit DIAG to efficiently solve such local quadratic problems to obtain improved convergence rates. Concretely, let

$$\widehat{f}(x; x_k) = \max_y g(x, y) + L\|x - x_k\|^2 . \tag{9}$$

By $L$-weak-convexity of $f$, $\widehat{f}(x; x_k)$ is *strongly*-convex–concave (Lemma 3) that can be solved using DIAG up to *certain accuracy* to obtain $x_{k+1}$. We refer to this algorithm as Prox-DIAG and provide a pseudo-code for the same in Algorithm 2. The following theorem gives convergence guarantees for

---

**Algorithm 2:** Proximal Dual Implicit Accelerated Gradient (Prox-DIAG) for nonconvex concave programming

---

    **Input:** $g, L, \varepsilon, x_0, y_0$
    **Output:** $x_k$
**1** Set $\tilde{\varepsilon} \leftarrow \frac{\varepsilon^2}{64 L}$
**2** **for** $k = 0, 1, \ldots, K$ **do**
**3**     Using DIAG for strongly convex concave minimax problem,

$$\min_x \max_{y \in \mathcal{Y}} \left[\widehat{g}(x, y; x_k) = g(x, y) + L\|x - x_k\|^2\right] \tag{10}$$

    find $x_{k+1}$ such that,

$$\max_{y \in \mathcal{Y}} g(x_{k+1}, y) + L\|x_{k+1} - x_k\|^2 \leq \min_x \max_{y \in \mathcal{Y}} g(x, y) + L\|x - x_k\|^2 + \frac{\tilde{\varepsilon}}{4} \tag{11}$$

    **if** $\max_{y \in \mathcal{Y}} g(x_k, y) - \frac{3\tilde{\varepsilon}}{4} \leq \max_{y \in \mathcal{Y}} g(x_{k+1}, y) + L\|x_{k+1} - x_k\|^2$ **then**
**4**         **return** $x_k$

---

Prox-DIAG.

**Theorem 2** (Convergence rate of Prox-DIAG). *Let $g(x, y)$ be $L$-smooth, $g(x, \cdot)$ be concave, $\mathcal{X}$ be $\mathbb{R}^p$, $\mathcal{Y}$ be a convex compact subset of $\mathbb{R}^q$, and the minimum value of function $f(x) = \max_{y \in \mathcal{Y}} g(x, y)$ be bounded below, i.e. $f(x) \geq f^* > -\infty$. Then Prox-DIAG (Algorithm 2) after,*

$$K = \left\lceil \frac{4^4 L(f(x_0) - f^*)}{3\varepsilon^2} \right\rceil$$

*steps outputs an $\varepsilon$-FOSP. The total first-order oracle complexity to output $\varepsilon$-FOSP is: $O\left(\frac{L^2 D_{\mathcal{Y}}(f(x_0) - f^*)}{\varepsilon^3} \log^2\left(\frac{1}{\varepsilon}\right)\right)$.*

A proof is provided in Appendix B.7. Note that Prox-DIAG solves the quadratic approximation problem to higher accuracy of $O(\epsilon^2)$ which then helps bounding the gradient of the Moreau envelope. Also due to the modular structure of the argument, a faster inner loop for special settings, e.g., when $g(x, y)$ is a finite-sum, can ensure more efficient algorithm. While our algorithm is able to significantly improve upon existing state-of-the-art rate of $O(1/\varepsilon^5)$ in general nonconvex-concave setting [18], it is unclear if the rate can be further improved. In fact, precise lower-bounds for this setting are mostly unexplored and we leave further investigation into lower-bounds as a topic of future research.

We also specialize the Prox-DIAG algorithm, as Prox-FDIAG (Algorithm 5 in Appendix C), for the case of minimizing a weakly convex $f(x)$, with the special structure of *finite max-type function*:

$$\min_x \left[f(x) = \max_{1 \leq i \leq m} f_i(x)\right] , \tag{P3}$$

where $f_i$'s could be nonconvex but are $L$-smooth, $G$-Lipschitz and bounded from below. For this case, we improve the current known best rate of $O\left(m/\varepsilon^4\right)$ and obtain a faster rate of $O(m\log^{3/2} m/\varepsilon^3)$ using the Prox-FDIAG algorithm. Please refer to Appendix C for more details.

## 5   Experiments

We empirically verify the performance of Prox-FDIAG (Algorithm 5 in Appendix C) on a synthetic finite max-type nonconvex minimization problem (P3). We consider the following problem. $\min_{x\in\mathbb{R}^2}\left[f(x)=\max_{1\leq i\leq m=9}f_i(x)\right]$ where $f_i(x)=q_{(-1,\,(X_i^{(1)},X_i^{(2)}),\,C_i)}(x)$ for all $1\leq i\leq 8$, where $q_{(a,b,c)}(x)=a\|x-b\|_2^2+c$, $X_i^{(1)}$, $X_i^{(2)}$, and $c_i$ are randomly generated. Thus each $f_i$ is smooth with parameter $L=1$, which implies that $f$ is $L$-weakly convex. We implement three algorithms: Prox-FDIAG (Algorithm 5, red circles), Adaptive Prox-FDIAG (Algorithm 6, black dots), and subgradient method [11] (blue triangles). Adaptive Prox-FDIAG is a practically faster variant of Prox-FDIAG, with the same first-order oracle complexity guarantee (up to an $O(\log(1/\varepsilon))$ factor). In Figure 1, we plot the norm of gradient of Moreau envelope $\|\nabla f_{\frac{1}{2L}}(x_k)\|_2$ against the number

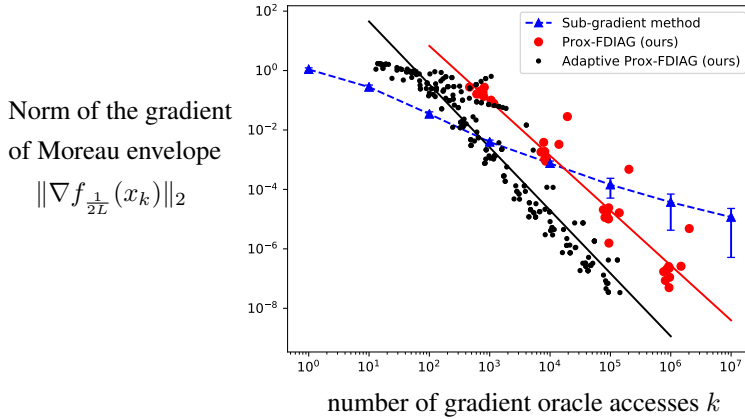

Figure 1: For small target accuracy $\varepsilon$ regime, Adaptive Prox-FDIAG (ours) has the fastest convergence rate followed by Prox-FDIAG (ours) and subgradient method.

of iterations $k$ in log-log scale. We see that, Prox-FDIAG and Adaptive Prox-FDIAG have a faster convergence rate than subgradient method, and Adaptive Prox-FDIAG is almost always faster than Prox-FDIAG. We provide more details about the algorithms and the experiments in Appendix E.

## 6   Conclusion

In this paper, we study smooth minimax problems, where the maximization is concave but the minimization is either strongly convex or nonconvex. In both of these settings, we present new algorithms improving state-of-the-art. The key ideas are i) a novel way to combine Mirror-Prox and Nesterov's AGD for strongly convex case that can tightly bound primal-dual gap and ii) an inexact prox method with good convergence rate to stationary points for the nonconvex case. While we only present our results for the Euclidean setting, generalizing it to non-Euclidean settings with the framework of Bregman divergences should be straight forward. Finally, we showcase the empirical superiority of our nonconvex algorithm over state-of-the-art subgradient method for a case of finite max-type nonconvex minimization problems. Some of the more interesting questions would be to understand the optimality of the rates that we obtain and dependence on the strong convexity parameter. Further extensions of these results to the stochastic setting would also be quite interesting.

## Acknowledgement

This work is partially supported by NSF awards CCF-1927712 and RI-1929955.

## Footnotes

[1]While [18] gives a rate of $O\left(1/k^{1/4}\right)$ with an approximate maximization oracle for $\max_{y \in \mathcal{Y}} g(x,y)$, taking into account the cost of implementing such a maximization oracle gives a rate of $O\left(1/k^{1/5}\right)$.

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
