[Supplementary Material]

# Appendix

## A  Nesterov's accelerated gradient descent

---
**Algorithm 3:** Nesterov's accelerated gradient ascent

---
**Input:** Smooth concave function $h(\cdot)$, learning rate $\frac{1}{\beta}$, initial points $y_0$ and $z_0$
**Output:** $y_k$
1 **for** $k = 0, 1, \ldots$ **do**
2  $\quad w_k \leftarrow (1 - \tau_k)y_k + \tau_k z_k, \ y_{k+1} \leftarrow \mathcal{P}_{\mathcal{Y}}\left(w_k + \frac{1}{\beta}\nabla h(w_k)\right),$
 $\quad\quad z_{k+1} \leftarrow \mathcal{P}_{\mathcal{Y}}\left(z_k + \eta_k \nabla h(w_k)\right)$

---

Nesterov's accelerated gradient descent [38] is an optimal method for minimizing smooth convex functions (or equivalently maximizing smooth concave functions). In order to simplify the exposition in the sequel, we will consider the algorithm for maximizing concave functions. The pseudocode for this is presented in Algorithm 3. Fix any point $y \in \mathcal{Y}$.

### A.1  Smooth concave function

Consider the potential function

$$\Phi(k) := k(k+1)\left(h(y) - h(y_k)\right) + 2\beta \cdot \|y - z_k\|^2.$$

The following lemma (from [2]) is the key result that helps us obtain the convergence rate of Algorithm 3. Here $\mathcal{P}_{\mathcal{Y}}(\cdot)$ denotes projection onto $\mathcal{Y}$.

**Lemma 2.** *[2] Suppose $h(\cdot)$ is an $L$-smooth concave function and the parameters of Algorithm 3 are chosen so that $\beta > L$, $\eta_k = \frac{k+1}{2\beta}$ and $\tau_k = \frac{2}{k+2}$. Then, we have*

$$\Phi(k+1) \leq \Phi(k).$$

*Proof of Lemma 2.*  Writing

$$\Phi(k+1) - \Phi(k) = (k+1)(k+2)\left(h(w_k) - h(y_{k+1})\right) \tag{12}$$
$$- k(k+1)\left(h(w_k) - h(y_k)\right) + 2(k+1)\left(h(y) - h(w_k)\right)$$
$$+ 2\beta\left(\|z_{k+1} - y\|^2 - \|z_k - y\|^2\right), \tag{13}$$

we bound the three terms appearing in separate lines above. Firstly, for the third term, $\|z_{k+1} - y\|^2 \leq \|z_k + \eta_k \nabla h(w_k) - y\|^2 - \|z_{k+1} - z_k - \eta_k \nabla h(w_k)\|^2$ due to Pythagoras theorem and so

$$\|z_{k+1} - y\|^2 - \|z_k - y\|^2 \leq 2\eta_k \langle \nabla h(w_k), z_k - y\rangle + \eta_k^2 \|\nabla h(w_k)\|^2 - \|z_{k+1} - z_k - \eta_k \nabla h(w_k)\|^2$$
$$\leq 2\eta_k \langle \nabla h(w_k), z_{k+1} - y\rangle - \|z_{k+1} - z_k\|^2. \tag{14}$$

For the second term, we have

$$- k(k+1)\left(h(w_k) - h(y_k)\right) + 2(k+1)\left(h(y) - h(w_k)\right)$$
$$\leq -k(k+1)\langle \nabla h(w_k), w_k - y_k\rangle + 2(k+1)\langle \nabla h(w_k), y - w_k\rangle = 2(k+1)\langle \nabla h(w_k), y - z_k\rangle \tag{15}$$

Finally, for the first term, we have $h(y_{k+1}) - h(w_k) \geq \langle \nabla h(w_k), y_{k+1} - w_k\rangle - \frac{\beta}{2}\|y_{k+1} - w_k\|^2$. Since $y_{k+1} = \operatorname{argmax}_{\bar{y} \in \mathcal{Y}} \langle \nabla h(w_k), \bar{y} - w_k\rangle - \frac{\beta}{2}\|\bar{y} - w_k\|^2$, we have for $v := (1-\tau_k)y_k + \tau_k z_{k+1} \in \mathcal{Y}$,

$$h(y_{k+1}) - h(w_k) \geq \langle \nabla h(w_k), y_{k+1} - w_k\rangle - \frac{\beta}{2}\|y_{k+1} - w_k\|^2$$

$$\geq \langle \nabla h(w_k), v - w_k\rangle - \frac{\beta}{2}\|v - w_k\|^2 = \tau_k \langle \nabla h(w_k), z_{k+1} - z_k\rangle - \frac{\beta\tau_k^2}{2}\|z_{k+1} - z_k\|^2, \tag{16}$$

where we used $w_k = (1 - \tau_k)y_k + \tau_k z_k$ in the last step. Substituting (16), (15) and (14) in (13) proves the lemma. $\qquad\square$

# B Proofs

## B.1 Auxiliary lemma

**Lemma 3.** *If $f(x)$ is a L-weakly convex function and $\tilde{f}(x)$ is a $\tilde{\sigma}(\geq L)$-strongly convex differentiable function, then $f(x) + \tilde{f}(x)$ is $(\tilde{\sigma} - L)$-strongly convex.*

*Proof.* Since $f$ is $L$-weakly convex and $\tilde{f}$ is $\sigma$-strongly convex we get that,

$$f(x') \geq f(x) + \langle u_x, x' - x \rangle - \frac{L}{2}\|x' - x\|^2,$$

$$\tilde{f}(x') \geq \tilde{f}(x) + \langle \nabla \tilde{f}(x), x' - x \rangle + \frac{\tilde{\sigma}}{2}\|x' - x\|^2,$$

$$\implies f(x') + \tilde{f}(x') \geq f(x) + \tilde{f}(x) + \langle u_x + \nabla \tilde{f}(x), x' - x \rangle + \frac{\tilde{\sigma} - L}{2}\|x' - x\|^2. \tag{17}$$

where $u_x \in \partial f(x)$. We finish the proof by noting that $\partial(f + \tilde{f}) = \partial f + \nabla \tilde{f}$ [27, Corollary 1.12.2.]. $\qquad\square$

## B.2 Properties of Moreau envelope

The following lemma provides some useful properties of the Moreau envelope for weakly convex functions.

**Lemma 4.** *For an $L$-weakly convex proper l.s.c. function $f : \mathcal{X} \to \mathbb{R} \cup \{\infty\}$ such that $\mathcal{X} = \mathbb{R}^p$ and $L < 1/\lambda$, the following hold true,*

(a) *The minimizer $\hat{x}_\lambda(x) = \arg\min_{x' \in \mathcal{X}} f(x') + \frac{1}{2\lambda}\|x - x'\|^2$ is unique and $f(\hat{x}_\lambda(x)) \leq f_\lambda(x) \leq f(x)$. Furthermore, $\arg\min_x f(x) = \arg\min_x f_\lambda(x)$.*

(b) *$f_\lambda$ is $\left(\frac{1}{\lambda} + \frac{1}{\lambda(1-\lambda L)}\right)$-smooth and thus differentiable, and*

(c) *$\min_{u \in \partial f(\hat{x}_\lambda(x))} \|u\| \leq (1/\lambda)\|\hat{x}_\lambda(x) - x\| = \|\nabla f_\lambda(x)\|$.*

*Proof.* We re-write $f_\lambda(x)$ as minimum value of a $(\frac{1}{\lambda} - L)$-strong convex function $\phi_{\lambda,x}$, as $f$ is $L$-weakly convex (Definition 3) and $\frac{1}{2\lambda}\|x - x'\|^2$ is differentiable and $\frac{1}{\lambda}$-strongly convex (Lemma 3),

$$f_\lambda(x) = \min_{x' \in \mathcal{X}} \left[ \phi_{\lambda,x}(x') = f(x') + \frac{1}{2\lambda}\|x - x'\|^2 \right]. \tag{18}$$

Then first part of (a) follows trivially by the strong convexity. For the second part notice the following,

$$\min_x f_\lambda(x) = \min_x \min_{x'} f(x') + \frac{1}{2\lambda}\|x - x'\|^2$$

$$= \min_{x'} \min_x f(x') + \frac{1}{2\lambda}\|x - x'\|^2$$

$$= \min_{x'} f(x')$$

Thus $\arg\min_x f_\lambda(x) = \arg\min_x f(x)$. For $(b)$ we can re-write the Moreau envelope $f_\lambda$ as,

$$f_\lambda(x) = \min_{x'} f(x') + \frac{1}{2\lambda}\|x - x'\|^2$$

$$= \frac{\|x\|^2}{2\lambda} - \frac{1}{\lambda}\max_{x'}(x^T x' - \lambda f(x') - \frac{\|x'\|^2}{2})$$

$$= \frac{\|x\|^2}{2\lambda} - \frac{1}{\lambda}\left(\lambda f(\cdot) + \frac{\|\cdot\|^2}{2}\right)^*(x) \tag{19}$$

where $(\cdot)^*$ is the Fenchel conjugation operator. Since $L < 1/\lambda$, using $L$-weak convexity of $f$, it is easy to see that $\lambda f(x') + \frac{\|x'\|^2}{2}$ is $(1 - \lambda L)$-strongly convex, therefore its Fenchel conjugate would

be $\frac{1}{(1-\lambda L)}$-smooth [21, Theorem 6]. This, along with $\frac{1}{\lambda}$-smoothness of first quadratic term implies that $f_\lambda(x)$ is $\left(\frac{1}{\lambda} + \frac{1}{\lambda(1-\lambda L)}\right)$-smooth, and thus differentiable.

For $(c)$ we again use the reformulation of $f_\lambda(x)$ as $\min_{x' \in \mathcal{X}} \phi_{\lambda,x}(x')$ (18). Then by first-order necessary condition for optimality of $\hat{x}_\lambda(x)$, we have that $x - \hat{x}_\lambda(x) \in \lambda \partial f(x)$. Further, from proof of part (a) we have that $\phi_{\lambda,x}(x')$ $(1 - \lambda L)$-strongly-convex in $x'$ and it is quadratic (and thus convex) in $x$. Then we can use Danskin's theorem [4, Section 6.11] to prove that, $\nabla f_\lambda(x) = (x - \hat{x}_\lambda(x))/\lambda \in \partial f(x)$. Refer [45, Section B.1] for other proofs of the same result.

$\square$

## B.3 Proof of Lemma 1

It is easy to see that $g(\cdot, y)$ is $L$-weakly convex if it is $L$-smooth: $g(x', y) \geq g(x, y) + \langle \nabla_x g(x, y), x' - x \rangle - \frac{L}{2}\|x' - x\|^2$. Thus we only need to prove the case of $L$-weakly convex $g(\cdot, y)$. Since $g(\cdot, y)$ is $L$-weakly convex we get that,

$$g(x', y) \geq g(x, y) + \langle u_{x,y}, x' - x \rangle - \frac{L}{2}\|x' - x\|^2$$

$$\implies g(x', y) + \frac{L}{2}\|x'\|^2 \geq g(x, y) + \frac{L}{2}\|x\|^2 + \langle u_{x,y} + Lx, x' - x \rangle$$

where $u_{x,y} \in \partial_x g(x, y)$. This means that $\tilde{g}(x, y) := g(x, y) + \frac{L}{2}\|x\|^2$ is convex, since $\partial_x \tilde{g}(x, y) = \partial_x g(x, y) + Lx$ [27, Corollary 1.12.2.].

Let $\tilde{f}(x) = \max_{y \in \mathcal{Y}} \tilde{g}(x, y)$. Since $\tilde{g}(x, y)$ is convex in $x$ and smooth (Definition 1), and $\mathcal{Y}$ is compact set we use Danskin's theorem [4, Section 6.11] to prove that,

$$\partial \tilde{f}(x) = \text{conv}\{\partial_x \tilde{g}(x, y^*) \mid y^* \in \arg\max_{y \in \mathcal{Y}} \tilde{g}(x, y)\},$$

$$\implies \partial f(x) + Lx = \text{conv}\{\partial_x g(x, y^*) + Lx \mid y^* \in \arg\max_{y \in \mathcal{Y}} g(x, y)\},$$

$$\implies \partial f(x) = \text{conv}\{\partial_x g(x, y^*) \mid y^* \in \arg\max_{y \in \mathcal{Y}} g(x, y)\}. \quad (20)$$

where the second to last step comes from the facts that $\partial \tilde{f} = \partial f + Lx$, $\partial_x \tilde{g}(x, y) = \partial_x g(x, y) + Lx$ [27, Corollary 1.12.2.], and $\arg\max_{y \in \mathcal{Y}} \tilde{g}(x, y) = \arg\max_{y \in \mathcal{Y}} g(x, y) + \frac{L}{2}\|x\|^2 = \arg\max_{y \in \mathcal{Y}} g(x, y)$. Let $u_{x,y} \in \partial_x g(x, y)$ and $y^* \arg\max_{y \in \mathcal{Y}} g(x, y)$ then,

$$f(x') \geq g(x', y^*) \overset{(a)}{\geq} g(x, y^*) + \langle u_{x,y^*}, x' - x \rangle - \frac{L}{2}\|x' - x\|^2$$

$$\overset{(b)}{\implies} f(x') \geq f(x) + \langle v_x, x' - x \rangle - \frac{L}{2}\|x' - x\|^2$$

where $(a)$ uses $L$-weak convexity of $g(\cdot, y)$, and $(b)$ uses (20) and $v_x \in \partial f(x)$.

## B.4 Pseudocode for Conceptual DIAG algorithm

The pseudocode for C-DIAG algorithm is presented in Algorithm 4.

## B.5 Proof of Theorem 1

A cursory glance of the DIAG (Algorithm 1) reveals that it is a modified version of projected accelerated gradient ascent (Algorithm 3) on some function of $y$ with a modified step given by `Imp-STEP`, which is inspired from the conceptual Mirror-Prox method of [34]. In the following lemma we analyze the `Imp-STEP` sub-routine, which is the most non-trivial step of the algorithm.

**Lemma 5.** *If $\beta = 2\frac{L^2}{\sigma}$, the sub-routine* `Imp-STEP`$(g, L, \sigma, w, \beta, \varepsilon_{\text{step}})$ *of Algorithm 1, returns a pair of points $(\hat{x}_R, y_{R+1}) \in \mathcal{X} \times \mathcal{Y}$, such that,*

$$g(\hat{x}_R, y_{R+1}) \leq \min_x g(x, y_R) + \varepsilon_{\text{step}}, \; and, \; y_R = \mathcal{P}_\mathcal{Y}\left(w + \frac{1}{\beta}\nabla_y g(\hat{x}_{R-1}, w)\right) \quad (21)$$

*in $R = \lceil \log_2\left((5LD_\mathcal{Y}/\sigma)\sqrt{L/2\varepsilon_{\text{step}}}\right)\rceil$ iterations with $O\left(\sqrt{L/\sigma}\log\left(1/\varepsilon_{\text{step}}\right)\right)$ gradient computations per iterations.*

---

**Algorithm 4:** Conceptual Dual Implicit Accelerated Gradient (C-DIAG) for strongly-convex–concave programming

---

**Input:** $g, L, \sigma, x_0, y_0, K$
**Output:** $\bar{x}_K, y_K$

1 Set $\beta \leftarrow 2\frac{L^2}{\sigma}$, $z_0 \leftarrow y_0$
2 **for** $k = 0, 1, \ldots, K - 1$ **do**
3 $\quad$ $\tau_k \leftarrow \frac{2}{(k+2)}$, $\eta_k \leftarrow \frac{(k+1)}{2\beta}$, $w_k \leftarrow (1 - \tau_k)y_k + \tau_k z_k$
4 $\quad$ Choose $x_{k+1}, y_{k+1}$ ensuring:

$$g(x_{k+1}, y_{k+1}) = \min_x g(x, y_{k+1}), \ \ y_{k+1} = \mathcal{P}_{\mathcal{Y}}\left(w_k + \frac{1}{\beta}\nabla_y g(x_{k+1}, w_k)\right)$$

5 $\quad$ $z_{k+1} \leftarrow \mathcal{P}_{\mathcal{Y}}\left(z_k + \eta_k \nabla_y g(x_{k+1}, w_k)\right), \ \ \bar{x}_{k+1} \leftarrow \frac{2}{(k+1)(k+2)}\sum_{i=1}^{k+1} i \cdot x_i$
6 **return** $\bar{x}_K, y_K$

---

A proof for this lemma is provided in Appendix B.5.1. The above lemma guarantees that the `Imp-STEP` sub-routine converges fast (linear time), in $O(\log(1/\varepsilon_{\text{step}}))$ steps with $O(\sqrt{L/\sigma}\log^2(1/\varepsilon_{\text{step}}))$ number of gradient computations.

In the rest of the proof we will utilize the recently proposed *potential-function* based proof for accelerated gradient decent (AGD) [2, Section 5.2]. Analyzing AGD using potential-function has an advantage over the standard analysis because, even though AGD does not decrease the function value monotonically the former constructs a potential-function which monotonically decreases over the iterations. Given the guarantees (Lemma 5) for the `Imp-STEP` sub-routine we can re-write an iteration of the DIAG algorithm by the following steps:

$$\tau_k = \frac{2}{(k+2)}, \ \ \eta_k = \frac{(k+1)}{2\beta} \tag{22}$$

$$w_k = (1 - \tau_k)y_k + \tau_k z_k \tag{23}$$

$$y_{k+1} = \mathcal{P}_{\mathcal{Y}}\left(w_k + \frac{1}{\beta}\nabla_y h_{x_{k+1}}(w_k)\right) \tag{24}$$

$$z_{k+1} = \mathcal{P}_{\mathcal{Y}}\left(z_k + \eta_k \nabla_y h_{x_{k+1}}(w_k)\right) \tag{25}$$

where $h_{k+1}(y) := g(x_{k+1}, y)$ such that $g(x_{k+1}, y_{k+1}) \leq \min_{x \in \mathcal{X}} g(x, y_{k+1}) + \varepsilon_{\text{step}}$. That is at iteration $k$, DIAG executes the $k$-th step of the accelerated gradient ascent for the concave function $h_{k+1} = g(x_{k+1}, \cdot)$ (Algorithm 3). As in (12), for the concave function $h_k : \mathcal{Y} \to \mathbb{R}$ and an arbitrary reference point $\tilde{y} \in \mathcal{Y}$, we define the following potential function for iteration $j$,

$$\Phi^{h_k}(j) = j(j+1)(h_k(\tilde{y}) - h_k(y_j)) + 2\beta\|z_j - \tilde{y}\|^2 \tag{26}$$

Since $g(x, \cdot)$ is $L$-smooth, it is also $\frac{2L^2}{\sigma}$-smooth ($\sigma \leq L$). Then, using Lemma 2, we see that for a step-size of $\frac{1}{\beta} = \frac{\sigma}{2L^2}$, the potential function $\Phi^{h_k}(k)$ decrease at step of $k$ of the algorithm:

$\Phi^{h_{k+1}}(k+1) \le \Phi^{h_{k+1}}(k)$. Thus,

$$
\begin{aligned}
\Phi^{h_{k+1}}(k+1) &\le \Phi^{h_{k+1}}(k) \\
&= k(k+1)(h_{k+1}(\tilde{y}) - h_{k+1}(y_k)) + 2\beta\|z_k - \tilde{y}\|^2 \\
&= k(k+1)(h_k(\tilde{y}) - h_k(y_k)) + 2\beta\|z_k - \tilde{y}\|^2 + \\
&\quad k(k+1)(h_{k+1}(\tilde{y}) - h_k(\tilde{y})) + k(k+1)(h_k(y_k) - h_{k+1}(y_k)) \\
&= \Phi^{h_k}(k) + k(k+1)(g(x_{k+1}, \tilde{y}) - g(x_k, \tilde{y})) + k(k+1)(g(x_k, y_k) - g(x_{k+1}, y_k)) \\
&\overset{(a)}{\le} \Phi^{h_k}(k) + k(k+1)(g(x_{k+1}, \tilde{y}) - g(x_k, \tilde{y})) + k(k+1)\varepsilon_{\text{step}}^{(k)} \quad (27)
\end{aligned}
$$

$$
\overset{(b)}{\implies} \Phi^{h_K}(K) \le \Phi^{h_0}(0) + \sum_{k=0}^{K-1} k(k+1)(g(x_{k+1}, \tilde{y}) - g(x_k, \tilde{y})) + \sum_{k=1}^{K-1} k(k+1)\varepsilon_{\text{step}}^{(k)}
$$

$$
\le \Phi^{h_0}(0) + (K-1)K g(x_K, \tilde{y}) - \sum_{k=1}^{K-1} 2k\, g(x_k, \tilde{y}) + \sum_{k=1}^{K-1} k(k+1)\varepsilon_{\text{step}}^{(k)} \quad (28)
$$

Where $(a)$ follows from Lemma 5 and $g(x_k, y_k) - g(x_{k+1}, y_k) \le g(x_k, y_k) - \min_x g(x, y_k) \le \varepsilon_{\text{step}}^{(k)}$, $(b)$ is obtained summing (27) over $k = \{0, \dots, K-1\}$. Rearranging the terms of (28) we get,

$$
\begin{aligned}
\Phi^{h_0}(0) + \sum_{k=1}^{K-1} k(k+1)\varepsilon_{\text{step}}^{(k)} &\ge \sum_{k=1}^{K-1} 2k\, g(x_k, \tilde{y}) + \Phi^{h_K}(K) - (K-1)K g(x_K, \tilde{y}) \\
&\ge \sum_{k=1}^{K-1} 2k\, g(x_k, \tilde{y}) + K(K+1)(g(x_K, \tilde{y}) - g(x_K, y_K)) + \\
&\quad 2\beta\|z_K - \tilde{y}\|^2 - (K-1)K g(x_K, \tilde{y}) \\
&\ge \sum_{k=1}^{K} 2k\, g(x_K, \tilde{y}) - K(K+1)g(x_K, y_K) \\
&\overset{(a)}{\ge} K(K+1)[g(\bar{x}_K, \tilde{y}) - g(x_K, y_K)] \\
&\overset{(b)}{\ge} K(K+1)[g(\bar{x}_K, \tilde{y}) - g(\tilde{x}, y_K) - \varepsilon_{\text{step}}^{(K)}] \quad (29)
\end{aligned}
$$

where $(a)$ uses the $\bar{x}_K = \frac{1}{K(K+1)} \sum_{k=1}^{K} (2i)\, x_i$ and convexity of $g(\cdot, \tilde{y})$, and $(b)$ uses Lemma 6. Thus we get that,

$$
\begin{aligned}
g(\bar{x}_K, \tilde{y}) - g(\tilde{x}, y_K) &\le \frac{\Phi^{h_0}(0)}{K(K+1)} + \sum_{k=1}^{K} \frac{k(k+1)}{K(K+1)} \varepsilon_{\text{step}}^{(k)} \\
&= \frac{2\beta\|y_0 - \tilde{y}\|^2}{K(K+1)} + \sum_{k=1}^{K} \frac{k(k+1)}{K(K+1)} \varepsilon_{\text{step}}^{(k)} \quad (30)
\end{aligned}
$$

Finally we get the desired general statement by taking minimum and maximum over $\tilde{x}$ and $\tilde{y}$ respectively. By selecting $\varepsilon_{\text{step}}^{(k)} = \frac{L^2 D_{\mathcal{Y}}^2}{\sigma k^3(k+1)}$ we get,

$$
\max_{\tilde{y} \in \mathcal{Y}} g(\bar{x}_K, \tilde{y}) - \min_{\tilde{x} \in \mathcal{X}} g(\tilde{x}, y_K) \le \frac{6 \frac{L^2}{\sigma} D_{\mathcal{Y}}^2}{K(K+1)} \quad (31)
$$

Further, using Lemma 5 and $\varepsilon_{\text{step}}^{(k)} = \frac{L^2 D_{\mathcal{Y}}^2}{\sigma k^3(k+1)}$, we get that the total number of gradient computations at iteration $k$ is at most $O\big(\sqrt{\frac{L}{\sigma}} \log^2(k)\big)$:

$$
\left\lceil \log_2 5k^2 \sqrt{\frac{L}{\sigma}} \right\rceil O\big(\sqrt{\frac{L}{\sigma}} \log\big(k^4\big)\big) \quad (32)
$$

Note that in updating $y_{k+1}$ in Eq. (24) and $x_{k+1}$ in `Imp-STEP` sub-routine, we were applying the principle of conceptual Mirror-Prox, where the update needs to satisfy some fixed point equation. This is critical in proving the above fast convergence rate.

### B.5.1 Proof of Lemma 5

For brevity, we define the following operations,

$$x^*(y) = \arg\min_{x \in \mathcal{X}} g(x, y) \tag{33}$$

$$y^+ = \mathcal{P}_\mathcal{Y}\left(w + \frac{1}{\beta}\nabla_y g(x^*(y), w)\right) \tag{34}$$

$x^*(y)$ is unique since $g(\cdot, y)$ is strongly convex. We first prove that, $x^*(y)$ is $\frac{L}{\sigma}$-Lipschitz continuous as follows.

$$
\begin{aligned}
\sigma \|x^*(y_2) - x^*(y_1)\|^2 &\overset{(a)}{\leq} \langle \nabla_x g(x^*(y_2), y_2) - \nabla_x g(x^*(y_1), y_2), x^*(y_2) - x^*(y_1)\rangle \\
&\overset{(b)}{\leq} \langle -\nabla_x g(x^*(y_1), y_2), x^*(y_2) - x^*(y_1)\rangle \\
&\overset{(c)}{\leq} \langle \nabla_x g(x^*(y_1), y_1) - \nabla_x g(x^*(y_1), y_2), x^*(y_2) - x^*(y_1)\rangle \\
&\overset{(d)}{\leq} L\|y_1 - y_2\|\|x^*(y_2) - x^*(y_1)\|
\end{aligned}
\tag{35}
$$

where $(a)$ uses $\sigma$-strong convexity of $g(\cdot, y)$, $(b)$ and $(c)$ use the necessary first order optimality conditions for $x^*(y_1)$ and $x^*(y_2)$: $\langle \nabla_x g(x^*(y), y), x - x^*(y)\rangle \geq 0$, and $(d)$ uses Cauchy-Schwarz inequality and $L$-smoothness of $g$ (Definition 1). Next we prove that the operation $(\cdot)^+$ is a contraction as follows,

$$
\begin{aligned}
\|y_1^+ - y_2^+\| &= \|\mathcal{P}_\mathcal{Y}\left(w + \frac{1}{\beta}\nabla_y g(x^*(y_1), w)\right) - \mathcal{P}_\mathcal{Y}\left(w + \frac{1}{\beta}\nabla_y g(x^*(y_2), w)\right)\| \\
&\overset{(a)}{\leq} \frac{1}{\beta}\|\nabla_y g(x^*(y_1), w) - \nabla_y g(x^*(y_2), w)\| \\
&\overset{(b)}{\leq} \frac{L}{\beta}\|x^*(y_1) - x^*(y_2)\| \\
&\overset{(c)}{\leq} \frac{L}{\beta}\frac{L}{\sigma}\|y_1 - y_2\| \overset{(d)}{\leq} 2^{-1}\|y_1 - y_2\|
\end{aligned}
\tag{36}
$$

where $(a)$ uses Pythagorean theorem and (34), $(b)$ uses $L$-smoothness of $g$, $(c)$ uses (35), and $(d)$ uses $\beta \geq 2\frac{LL}{\sigma}$. Therefore, $(\cdot)^+$ is a contraction by Banach's fixed point theorem, and thus it has a unique fixed point $\tilde{y}$: $(\tilde{y})^+ = \tilde{y}$, as $\mathcal{Y}$ is a compact (and hence complete) metric space. Now we will prove that the output of Imp-STEP, $(\hat{x}_R, y_{R+1})$ satisfies (21). Notice that if $\varepsilon_{\text{agd}}$ is small then $\hat{x}_r$ is close to $x^*(y_r)$:

$$
\frac{\sigma}{2}\|\hat{x}_r - x^*(y_r)\|^2 \overset{(a)}{\leq} g(\hat{x}_r, y_r) - \min_x g(x, y_r) \overset{(b)}{\Longrightarrow} \|\hat{x}_r - x^*(y_r)\| \leq \sqrt{\frac{2\varepsilon_{\text{agd}}}{\sigma}} = \frac{\beta\varepsilon_{\text{mp}}}{4L} \tag{37}
$$

where $(a)$ uses $\sigma$-strong convexity and optimality of $x^*(y_r)$, and $(b)$ uses (7), and $(c)$ uses $\varepsilon_{\text{agd}} = \sigma\beta^2\varepsilon_{\text{mp}}/(32L^2)$. Next we see that $\|y_r - \tilde{y}\|$ decreases to $\varepsilon$ exponentially fast.

$$
\begin{aligned}
\|y_r - \tilde{y}\| &\overset{(a)}{=} \|\mathcal{P}_\mathcal{Y}\left(w + \frac{1}{\beta}\nabla_y g(\hat{x}_{r-1}, w)\right) - (\tilde{y})^+\| \\
&\overset{(b)}{\leq} \|y_{r-1}^+ - (\tilde{y})^+\| + \|\mathcal{P}_\mathcal{Y}\left(w + \frac{1}{\beta}\nabla_y g(x^*(y_{r-1}), w)\right) - \mathcal{P}_\mathcal{Y}\left(w + \frac{1}{\beta}\nabla_y g(\hat{x}_{r-1}, w)\right)\| \\
&\overset{(c)}{\leq} 2^{-1}\|y_{r-1} - \tilde{y}\| + \frac{L}{\beta}\|x^*(y_{r-1}) - \hat{x}_{r-1}\| \\
&\overset{(d)}{\leq} 2^{-1}\|y_{r-1} - \tilde{y}\| + \frac{\varepsilon_{\text{mp}}}{4} \tag{38} \\
&\overset{(e)}{\leq} 2^{-r}\|y_0 - \tilde{y}\| + \frac{\varepsilon_{\text{mp}}}{2} \tag{39}
\end{aligned}
$$

where $(a)$ uses $y_{r+1} = \mathcal{P}_\mathcal{Y}\left(w + \frac{1}{\beta}\nabla_y g(\hat{x}_r, w)\right)$ and the fact that $\tilde{y} = (\tilde{y})^+$ is a fixed point, $(b)$ uses triangular inequality and (34), $(c)$ uses (36), Pythagorean theorem and $L$-smoothness of $g$

(Definition 1), $(d)$ uses (37), and $(e)$ just unrolls the recurrence relation in (38) . Next, we prove that the minimizer at $y_{R+1}$, $x^*(y_{R+1})$ is not far from $\hat{x}_R$.

$$\|x^*(y_{R+1}) - \hat{x}_R\| \overset{(a)}{\leq} \|x^*(y_{R+1}) - x^*(\tilde{y})\| + \|x^*(\tilde{y}) - x^*(y_R)\| + \|x^*(y_R) - \hat{x}_R\|$$

$$\overset{(b)}{\leq} \frac{L}{\sigma}(\|y_{R+1} - \tilde{y}\| + \|y_R - \tilde{y}\|) + \frac{\beta\varepsilon_{\mathrm{mp}}}{4L}$$

$$\overset{(c)}{\leq} \frac{L}{\sigma}(\varepsilon_{\mathrm{mp}} + \varepsilon_{\mathrm{mp}}) + \frac{\beta\varepsilon_{\mathrm{mp}}}{4L} = (2\frac{L}{\sigma} + \frac{\beta}{4L})\varepsilon_{\mathrm{mp}} \qquad (40)$$

where $(a)$ uses triangle inequality, and $(b)$ uses (35) and 37, and $(c)$ uses (39) and the fact that $R = \lceil \log_2 \frac{2D_{\mathcal{Y}}}{\varepsilon_{\mathrm{mp}}} \rceil$. Finally, we prove that $(x_R, y_{R+1})$ satisfies (21).

$$g(\hat{x}_R, y_{R+1}) \overset{(a)}{\leq} g(x^*(y_{R+1}), y_{R+1}) + \langle \nabla_x g(x^*(y_{R+1}), y_{R+1}), \hat{x}_R - x^*(y_{R+1}), \rangle + \frac{L}{2}\|x^*(y_{R+1}) - \hat{x}_R\|^2$$

$$\overset{(b)}{\leq} \min_x g(x, y_{R+1}) + 0 + \frac{25LL^2\varepsilon_{\mathrm{mp}}^2}{8\sigma^2} \overset{(c)}{=} \min_x g(x, y_{R+1}) + \varepsilon_{\mathrm{step}} \qquad (41)$$

where $(a)$ uses $L$-smoothness of $g(\cdot, y)$, $(b)$ uses necessary first order optimality condition: $\langle \nabla_x g(x^*(y), y), x - x^*(y) \rangle = 0$ and (40), and $(c)$ uses $\varepsilon_{\mathrm{mp}} = \frac{2\sigma}{5L}\sqrt{\frac{2\varepsilon_{\mathrm{step}}}{L}}$.

Let the number of gradient computations done per iteration of `Imp-STEP` (a run of accelerated gradient ascent) be $T_r$ and $\kappa = \sqrt{L/\sigma}$. Then, from guarantee on AGD ([2, Eqn. (5.68)]), we get that,

$$g(\hat{x}_r, y_r) - g(x^*(y_r), y_r) \leq \left(1 + \frac{1}{\sqrt{\kappa} - 1}\right)^{-T_r}\left(g(x_0, y_r) - g(x^*(y_r), y_r) + \frac{\sigma}{2}\|x_0 - x^*(y_r)\|^2\right)$$

$$\leq e^{-T_r/\sqrt{\kappa}} 2\left(g(x_0, y_r) - g(x^*(y_r), y_r)\right)$$

$$\leq e^{-T_r/\sqrt{\kappa}} 2\left(f(x_0) - h(y_r)\right)$$

$$\leq e^{-T_r/\sqrt{\kappa}} 2\left(f(x_0) - \min_{y' \in D_{\mathcal{Y}}} h(y')\right), \qquad (42)$$

where $\min_{y' \in D_{\mathcal{Y}}} h(y')$ is well-defined since $\mathcal{Y}$ is compact and $h$ is smooth (Lemma 6). This means that if we want $g(\hat{x}_r, y_r) - g(x^*(y_r), y_r) \leq \varepsilon_{\mathrm{agd}}$, then required number of steps $T_r$ is at most,

$$\left\lceil \sqrt{\frac{L}{\sigma}} \log \frac{2(f(x_0) - \min_{y' \in D_{\mathcal{Y}}} h(y'))}{\varepsilon_{\mathrm{agd}}} \right\rceil = \left\lceil \sqrt{\frac{L}{\sigma}} \log \frac{50L(f(x_0) - \min_{y' \in D_{\mathcal{Y}}} h(y'))}{\sigma\varepsilon_{\mathrm{step}}} \right\rceil$$

$$= O\left(\sqrt{\frac{L}{\sigma}} \log\left(\frac{1}{\varepsilon_{\mathrm{step}}}\right)\right) \qquad (43)$$

### B.6 Smoothness of dual of strongly-convex–concave minimax problem

**Lemma 6.** *For a $\sigma$-strongly-convex–concave $L$-smooth function $g(\cdot, \cdot)$, $h(u) = \min_{x \in \mathcal{X}} g(x, u)$ is an $\left(L + \frac{L^2}{\sigma}\right)$-smooth concave function.*

*Proof.* We know that $h(y) = \min_{x \in \mathcal{X}} g(x, y)$, where $g(\cdot, y)$ is $\sigma$-strongly convex, $g(x, \cdot)$ is concave, $g$ is $L$-smooth (Definition 1). Since $g(\cdot, y)$ is strongly convex, the minimizer $x^*(y) = \arg\min_{x \in \mathcal{X}} g(x, y)$ unique. Then by Danskin's theorem [4, Section 6.11], $h$ is differentiable and $\nabla h(y) = \nabla_y g(x^*(y), y)$. Then $h$ can be show to be smooth as follows,

$$\|\nabla h(y_1) - \nabla h(y_1)\| = \|\nabla_y g(x^*(y_1), y_1) - \nabla_y g(x^*(y_2), y_2)\|$$

$$\leq \|\nabla_y g(x^*(y_1), y_1) - \nabla_y g(x^*(y_1), y_2)\| + \|\nabla_y g(x^*(y_1), y_2) - \nabla_y g(x^*(y_2), y_2)\|$$

$$\overset{(a)}{\leq} L\|y_1 - y_2\| + L\|x^*(y_1) - x^*(y_2)\|$$

$$\overset{(b)}{\leq} L\|y_1 - y_2\| + L\frac{L}{\sigma}\|y_1 - y_2\| = \left(L + \frac{LL}{\sigma}\right)\|y_1 - y_2\| \qquad (44)$$

where $(a)$ uses $L$-smoothness of $g$ and $(b)$ uses (35). $\qquad \square$

## B.7 Proof of Theorem 2

We first note that by Lemma 3 and $L$-weak convexity of $g(\cdot, y)$ and $2L$-strong convexity of $L\|x - x_k\|^2$, $\widehat{g}(x, y; x_k) := g(x, y) + L\|x - x_k\|^2$ is $L$-strongly-convex. Similarly, $\widehat{f}(\cdot; x_k) := \max_{y \in \mathcal{Y}}[\widehat{g}(x, y; x_k) = g(x, y) + L\|x - x_k\|^2]$ is also $L$-strongly-convex.

We now divide the analysis of each iteration of our algorithm into two cases:

**Case 1:** $\widehat{f}(x_{k+1}; x_k) \leq f(x_k) - 3\tilde{\varepsilon}/4$. As every instance of Case 1 ensures $f(x_{k+1}) \leq \widehat{f}(x_{k+1}; x_k) \leq f(x_k) - 3\tilde{\varepsilon}/4$, we can have only $\left\lceil \frac{4(f(x_0) - f^*)}{3\tilde{\varepsilon}} \right\rceil$ Case 1 steps before termination. This claim requires monotonic decrease in $f(x_k)$ which holds until $f(x_{k+1}) \geq f(x_k)$, after which $\widehat{f}(x_{k+1}; x_k) \geq f(x_k)$, which in-turn imply that Prox-DIAG terminates (see termination condition of Prox-DIAG).

**Case 2:** $\widehat{f}(x_{k+1}; x_k) > f(x_k) - 3\tilde{\varepsilon}/4$: In this case, we show that $x_k$ is already an $\varepsilon$-FOSP and the algorithm returns $x_k$.

$$f(x_k) - \frac{3\tilde{\varepsilon}}{4} < \widehat{f}(x_{k+1}; x_k) \leq \min_x \widehat{f}(x; x_k) + \frac{\tilde{\varepsilon}}{4} \implies f(x_k) < \min_x \widehat{f}(x; x_k) + \tilde{\varepsilon} \quad (45)$$

Define $x_k^*$ as the point satisfying $x_k^* = \arg\min_x \widehat{f}(x; x_k)$. By $L$-strong convexity of $\widehat{f}(\cdot; x_k)$ (9), we prove that $x_k$ is close to $x_k^*$:

$$\widehat{f}(x_k^*; x_k) + \frac{L}{2}\|x_k - x_k^*\|^2 \leq \widehat{f}(x_k; x_k) = f(x_k) \overset{(a)}{<} \widehat{f}(x_k^*; x_k) + \tilde{\varepsilon} \implies \|x_k - x_k^*\| < \sqrt{\frac{2\tilde{\varepsilon}}{L}} \quad (46)$$

where $(a)$ uses (45). Now consider any $\tilde{x} \in \mathcal{X}$, such that $4\sqrt{\tilde{\varepsilon}/L} \leq \|\tilde{x} - x_k\|$. Then,

$$f(\tilde{x}) + L\|\tilde{x} - x_k\|^2 = \max_{y \in \mathcal{Y}} g(\tilde{x}, y) + L\|\tilde{x} - x_k\|^2 = \widehat{f}(\tilde{x}; x_k) \overset{(a)}{\geq} \widehat{f}(x_k^*; x_k) + \frac{L}{2}\|\tilde{x} - x_k^*\|^2$$

$$\overset{(b)}{\geq} f(x_k) - \tilde{\varepsilon} + \frac{L}{2}(\|\tilde{x} - x_k\| - \|x_k - x_k^*\|)^2 \overset{(c)}{\geq} f(x_k) + \tilde{\varepsilon}, \quad (47)$$

where $(a)$ uses uses $L$-strong convexity of $\widehat{f}(\cdot; x_k)$ at its minimizer $x_k^*$, $(b)$ uses (45), and $(b)$ and $(c)$ use triangle inequality, (46) and $4\sqrt{\tilde{\varepsilon}/L} \leq \|\tilde{x} - x_k\|$.

Now consider the Moreau envelope, $f_{\frac{1}{2L}}(x) = \min_{x' \in X} \phi_{\frac{1}{2L}, x}(x')$ where $\phi_{\lambda, x}(x') = f(x') + L\|x - x'\|^2$. Then, we can see that $\phi_{\frac{1}{2L}, x_k}(x')$ achieves its minimum in the ball $\{x' \in \mathcal{X} \mid \|x' - x_k\| \leq 4\sqrt{\tilde{\varepsilon}/L}\}$ by (47) and Lemma 4(a). Then, with Lemma 4(b,c) and $\tilde{\varepsilon} = \frac{\varepsilon^2}{64L}$, we get that,

$$\|\nabla f_{\frac{1}{2L}}(x_k)\| \leq (2L)\|x_k - \hat{x}_{\frac{1}{2L}}(x_k)\| = 8\sqrt{L\tilde{\varepsilon}} = \varepsilon, \quad (48)$$

i.e., $x_k$ is an $\varepsilon$-FOSP.

By combining the above two cases, we establish that $O\left(\left\lceil \frac{4(f(x_0) - f^*)}{3\tilde{\varepsilon}} \right\rceil\right)$ "outer" iterations ensure convergence to a $\varepsilon$-FOSP. We now compute the first-order complexity of each of these "outer" iterations. Recall that we use use the DIAG (Algorithm 1) algorithm for $L$-strongly-convex concave $2L$-smooth minimax problem to solve the inner optimization problem. So, if for each iteration of inner problem, DIAG algorithm takes $K$ steps then, by $\tilde{\varepsilon} = \frac{\varepsilon^2}{64L}$ and Theorem 1,

$$\frac{6(2L)^2 D_{\mathcal{Y}}^2}{LK^2} \leq \frac{\tilde{\varepsilon}}{4} = \frac{\varepsilon^2}{2^8 L} \implies O\left(\frac{LD_{\mathcal{Y}}}{\varepsilon}\right) \leq K \quad (49)$$

Therefore the number of gradient computations required for each iteration of inner problem is $O\left(\frac{LD_{\mathcal{Y}}}{\epsilon} \log^2\left(\frac{1}{\varepsilon}\right)\right)$ (Theorem 1), which along with the bound on the number of outer iterations establishes the Theorem's upper bound on the number of first-order oracle calls.

## C Minimizing finite max-type function with smooth components

As a special case of nonconvex–concave minimax problem, consider minimizing a weakly convex $f(x)$, with a special structure of *finite max-type function*:

$$\min_x \left[ f(x) = \max_{1 \leq i \leq m} f_i(x) \right], \quad (P3)$$

where $x \in \mathbb{R}^p$, the functional components $f_i(x)$'s could be *nonconvex* but are $L$-smooth and $G$-Lipschitz. Suppose $f$ itself takes a minimum value $f^* > -\infty$. For this problem, we propose and study a proximal (Prox-FDIAG) algorithm (Algorithm 5 presented in Appendix C.1) that is inspired by Algorithm 2 with the inner problem-solver replaced by Nesterov's finite convex minimax scheme [37, Section 2.3.1] instead of Algorithm 1. Using same proof technique as Theorem 2, we get:

**Corollary 1** (Convergence rate of Prox-FDIAG). *If the functional components $f_i(x)$'s are $G$-Lipschitz and $L$-smooth, and the optimal solution is bounded below, i.e. $f(x) \geq f^* > -\infty$, then after: $K = \left\lceil \frac{4^4 L(f(x_0) - f^*)}{3\varepsilon^2} \right\rceil$ outer steps, Prox-FDIAG outputs an $\varepsilon$-FOSP. The total first-order oracle complexity to find $\varepsilon$-FOSP is: $\left\lceil \frac{4^4 L(f(x_0) - f^*)}{3\varepsilon^2} \right\rceil \cdot \left\lceil \frac{2^4 G}{\varepsilon}(m \log^{3/2} m) \right\rceil$.*

See Appendix C.1 for a proof. Current best rate for this problem is achieved by subgradient methods. As the subgradient of a finite minimax function $\nabla_{i^*} f(x)$ is easy to evaluate, where $i^* \in \arg\max_i f_i(x)$, a rate of $O(m/\varepsilon^4)$ first-order oracle and function calls is achieved by the state-of-the-art subgradient method in [11]. We can obtain a similar result using Algorithm 1 but it requires extension to non-Euclidean settings with the framework of Bregman divergences. This is fairly standard and will be updated in the next version of the paper.

---

**Algorithm 5:** Proximal Finite Dual Implicit Accelerated Gradient (Prox-FDIAG) for finite nonconvex concave minimax optimization

> **Input:** functional components $\{f_i\}_{i=1}^m$, Lipschitzness $G$, smoothness $L$, domain $\mathcal{X}$, target accuracy $\varepsilon$, initial point $x_0$
>
> **Output:** $x_k$

**1** $\tilde{\varepsilon} \leftarrow \frac{\varepsilon^2}{64 L}$

**2 for** $k = 0, 1, \ldots$ **do**

**3** $\quad$ Using excessive gap technique [35, Problem (7.11)] for strongly convex components, find $x_{k+1} \in \mathcal{X}$ such that,

$$\widehat{f}(x_{k+1}; x_k) \leq \min_x \widehat{f}(x; x_k) + \tilde{\varepsilon}/4 \qquad (50)$$

$\quad$ **if** $f(x_k) - 3\tilde{\varepsilon}/4 < \widehat{f}(x_{k+1}; x_k)$ **then**

**4** $\quad\quad$ **return** $x_k$

---

### C.1 Proof of Corollary 1

Let

$$\widehat{f}(x; x_k) = \max_{1 \leq i \leq m} f_i(x_k) + \langle \nabla f_i(x_k), x - x_k \rangle + \frac{L}{2}\|x - x_k\|^2 \qquad (51)$$

be a quadratic approximation of the finite max-type function $f(x)$ at $x_k$. Then, $\widehat{f}(\cdot; x_k)$ is $L$-strongly convex, since it is a maximum of convex functions and the quadratic term in (51) is independent of $i$.

Proof is similar to that of Theorem 2. We divide the analysis of each iteration of our algorithm into two cases.

**Case 1:** $\widehat{f}(x_{k+1}; x_k) \leq f(x_k) - 3\tilde{\varepsilon}/4$**.** This ensure that at iteration $k$ the objective value decreases by at least $3\tilde{\varepsilon}/4$ since, $f(x_{k+1}) \leq \widehat{f}(x_{k+1}; x_k) \leq f(x_k) - 3\tilde{\varepsilon}/4$. One cannot have more than $\left\lceil \frac{4(f(x_0) - f^*)}{3\tilde{\varepsilon}} \right\rceil$ instances of Case 1, before termination.

**Case 2:** $\widehat{f}(x_{k+1}; x_k) > f(x_k) - 3\tilde{\varepsilon}/4$**:** We show that $x_k$ is an $\varepsilon$-FOSP as follows.

$$f(x_k) - \frac{3\tilde{\varepsilon}}{4} < \widehat{f}(x_{k+1}; x_k) \leq \min_x \widehat{f}(x; x_k) + \frac{\tilde{\varepsilon}}{4} \implies f(x_k) < \min_x \widehat{f}(x; x_k) + \tilde{\varepsilon} \quad (52)$$

Define $x_k^*$ as the point satisfying $x_k^* = \arg\min_x \widehat{f}(x; x_k)$. By $L$-strong convexity of $\widehat{f}(\cdot, x_k)$ (51), we prove that $x_k$ is close to $x_k^*$:

$$\widehat{f}(x_k^*; x_k) + \frac{L}{2}\|x_k - x_k^*\|^2 \;\le\; \widehat{f}(x_k; x_k) \;=\; f(x_k) \;\overset{(a)}{<}\; \widehat{f}(x_k^*; x_k) + \tilde{\varepsilon}$$

$$\implies \; \|x_k - x_k^*\| < \sqrt{\frac{2\tilde{\varepsilon}}{L}} \tag{53}$$

where $(a)$ uses (52). Now consider any $\tilde{x} \in \mathcal{X}$, such that $4\sqrt{\tilde{\varepsilon}/L} \le \|\tilde{x} - x_k\|$. Then,

$$\begin{aligned}
f(\tilde{x}) + L\|\tilde{x} - x_k\|^2 &= \max_i f_i(\tilde{x}) + L\|\tilde{x} - x_k\|^2 \\[4pt]
&\overset{(a)}{\ge} \max_i f_i(\tilde{x}) + \langle \nabla f_i(x_k), \tilde{x} - x_k \rangle + \frac{L}{2}\|\tilde{x} - x_k\|^2 \\[4pt]
&\overset{(b)}{=} \widehat{f}(\tilde{x}; x_k) \\[4pt]
&\overset{(c)}{\ge} \widehat{f}(x_k^*; x_k) + \frac{L}{2}\|\tilde{x} - x_k^*\|^2 \\[4pt]
&\overset{(d)}{\ge} f(x_k) - \tilde{\varepsilon} + \frac{L}{2}(\|\tilde{x} - x_k\| - \|x_k - x_k^*\|)^2 \\[4pt]
&\overset{(e)}{\ge} f(x_k) - \tilde{\varepsilon} + 2\tilde{\varepsilon} \;=\; f(x_k) + \tilde{\varepsilon}
\end{aligned} \tag{54}$$

where $(a)$ uses weak convexity of $f_i$, $(b)$ uses (51), $(c)$ uses $L$-strong convexity of $\widehat{f}(\cdot; x_k)$ at its minimizer $x_k^*$, $(d)$ uses (52), and $(b)$ and $(e)$ use triangle inequality, (53) and $4\sqrt{\tilde{\varepsilon}/L} \le \|\tilde{x} - x_k\|$.

Now consider the Moreau envelope, $f_{\frac{1}{2L}}(x) = \min_{x' \in X} \phi_{\frac{1}{2L}, x}(x')$ where $\phi_{\lambda, x}(x') = f(x') + L\|x - x'\|^2$. Then, we can see that $\phi_{\frac{1}{2L}, x_k}(x')$ achieves its minimum in the ball $\{x' \in \mathcal{X} \,|\, \|x' - x_k\| \le 4\sqrt{\tilde{\varepsilon}/L}\}$ by (54) and Lemma 4(a). Thus, with Lemma 4(b,c), we get that,

$$\|\nabla f_{\frac{1}{2L}}(x_k)\| \le (2L)\|x_k - \hat{x}_{1/2L}(x_k)\| = 8\sqrt{L\tilde{\varepsilon}} = \varepsilon \tag{55}$$

Now we use the excessive gap technique for non-smooth strongly convex functions with max-structure to solve the inner optimization problem in $4G(m\log m)\sqrt{\frac{\log m}{\tilde{\varepsilon}L}}$ computations [35, Problem (7.11)].

Putting these together we see that the total number of inner steps to reach $\varepsilon$-FOSP is,

$$\left\lceil \frac{4(f(x_0) - f^*)}{3\tilde{\varepsilon}} \right\rceil \left\lceil 2G(m\log m)\sqrt{\frac{\log m}{L\tilde{\varepsilon}}} \right\rceil = \left\lceil \frac{4^4 L(f(x_0) - f^*)}{3\varepsilon^2} \right\rceil \left\lceil \frac{2^5 G}{\varepsilon}(m\log^{3/2} m) \right\rceil \tag{56}$$

## C.2  Adaptive Prox-FDIAG algorithm

In this section, we provide the Adaptive Prox-FDIAG (Algorithm 6) to find an $\varepsilon$-FOSP of the finite max-type nonconvex minimax problem P3 with $L$-smooth components. Adaptive Prox-FDIAG is a variation of the Prox-FDIAG (Algorithm 5). Adaptive Prox-FDIAG uses Prox-FDIAG as a sub-routine and successively finds $\varepsilon'$-FOSPs, for geometrically decreasing values of $\varepsilon'$ starting from $\varepsilon_0$ ($\ge \varepsilon$) until $\varepsilon'$ becomes equal to $\varepsilon$. It uses the $\varepsilon'$-FOSP as the starting point to find an $\varepsilon'/2$-FOSP. In the following corollary, we show that Adaptive Prox-FDIAG has the same the first-order oracle complexity (up to a $O(\log(\frac{1}{\varepsilon}))$ factor) as the Prox-FDIAG.

**Corollary 2** (Convergence rate of Adaptive Prox-FDIAG). *If the functional components $f_i(x)$'s are $G$-Lipschitz and $L$-smooth, and the optimal solution is bounded below, i.e. $f(x) \ge f^* > -\infty$, then after: $K = \left\lceil \log_2 \frac{\varepsilon_0}{\varepsilon} \right\rceil$ outer steps, Adaptive Prox-FDIAG outputs an $\varepsilon$-FOSP. The total first-order oracle complexity to find $\varepsilon$-FOSP is: $\left\lceil \log_2 \frac{\varepsilon_0}{\varepsilon} \right\rceil \left\lceil \frac{4^4 L(f(x_0) - f^*)}{3\varepsilon^2} \right\rceil \cdot \left\lceil \frac{2^4 G}{\varepsilon}(m\log^{3/2} m) \right\rceil$.*

*Proof.* Notice that, each iteration of Adaptive Prox-FDIAG for finding an $\varepsilon'$-FOSP, is a run of Prox-FDIAG (Algorithm 5), which has a maximum first-order oracle complexity of $\left\lceil \frac{4^4 L(f(x_0) - f^*)}{3\varepsilon^2} \right\rceil$.

$\left\lceil \frac{2^4 G}{\varepsilon} (m \log^{3/2} m) \right\rceil$ for finding an $\varepsilon'$-FOSP (Corollary 1), as $\varepsilon \leq \varepsilon'$. Further, since $\varepsilon'$ starts at $\varepsilon_0$ and halves after each iteration until $\varepsilon'$ becomes less than or equal to $\varepsilon$, the total number of outer iterations is $K = \left\lceil \log_2 \frac{\varepsilon_0}{\varepsilon} \right\rceil$. □

Therefore, Adaptive Prox-FDIAG has the same first-order oracle complexity as Prox-FDIAG, up to a $O(\log(\frac{1}{\varepsilon}))$ factor. However, we observe that Adaptive Prox-FDIAG converges faster than Prox-FDIAG in our experiments.

---

**Algorithm 6:** Adaptive Proximal Finite Dual Implicit Accelerated Gradient (Adaptive Prox-FDIAG) for finite nonconvex concave minimax optimization

---

**Input:** functional components $\{f_i\}_{i=1}^m$, Lipschitzness $G$, smoothness $L$, domain $\mathcal{X}$, target accuracy $\varepsilon$, initial point $x_0$, initial accuracy $\varepsilon_0$

**Output:** $x_k$

1   $\varepsilon' \leftarrow \max(\varepsilon_0, \ \varepsilon)$
2   **for** $k = 0, 1, \ldots$ **do**
3      Using Prox-FDIAG (Algorithm 5) initialized at $x_k$, find $x_{k+1} \in \mathcal{X}$ such that $x_{k+1}$ is an $\varepsilon'$-FOSP (Definition 6) of the function $f(x) = \max_{1 \leq i \leq m} f_i(x)$
4      **if** $\varepsilon = \varepsilon'$ **then**
5          $k \leftarrow k + 1$
6          **return** $x_k$
7      **else**
8          $\varepsilon' \leftarrow \max(\frac{\varepsilon'}{2}, \ \varepsilon)$

---

## D   Smoothing technique for strongly-convex–concave minimax problem

In this section we propose and analyze a smoothing technique [36] based indirect algorithm for solving the $L$-smooth $\sigma$-strongly-convex–concave minimax problem. The basic idea is to solve a smoothed (perturbed) version of the original function, $\tilde{g}(x, y) = g(x, y) - \varepsilon \|y\|^2 / 2 D_{\mathcal{Y}}^2$, which would be a strongly-convex–strongly-concave minimax problem. [1] proposes solving a strongly-convex–strongly-concave problem in linear rate using inexact accelerated gradient descent on its dual, whose main guarantee is given in the theorem below.

**Theorem 3.** *[1] Inexact accelerated gradient ascent on the dual problem can find an $\varepsilon$-primal dual pair of an $L$-smooth $\sigma_x$-strongly-convex–$\sigma_y$-strongly-concave problem:* $\min_x \max_y g(x, y)$*, with*
$\widetilde{O}\left( \sqrt{\frac{L + \frac{L^2}{\sigma_x}}{\sigma_y}} \sqrt{\frac{L}{\sigma_x}} \right)$ *calls to the first order gradient oracle of $g$.*

Now using this algorithm on the function $\tilde{g}$ can recover the same rate as DIAG method as follows. Plugging in $L = O(L)$, $\sigma_x = \sigma$, and $\sigma_y = \varepsilon / D_{\mathcal{Y}}^2$ into the algorithm complexity of Theorem 3 gives you a complexity of,

$$\widetilde{O}\left( \frac{L D_{\mathcal{Y}}}{\sqrt{\sigma \varepsilon}} \sqrt{\frac{L}{\sigma}} \right),$$

finding an $\varepsilon$-primal dual pair, $(\bar{x}, \bar{y})$, of $\tilde{g}$. Since $\max_{y \in \mathcal{Y}} g(\bar{x}, y) \leq \max_{y \in \mathcal{Y}} \tilde{g}(\bar{x}, y) + \varepsilon/2$ and $\tilde{g}(x, \bar{y}) \leq g(x, \bar{y})$, we get that,

$$\max_{y \in \mathcal{Y}} g(\bar{x}, y) - \min_{x \in \mathcal{X}} g(x, \bar{y}) \leq \max_{y \in \mathcal{Y}} \tilde{g}(\bar{x}, y) - \min_{x \in \mathcal{X}} \tilde{g}(x, \bar{y}) + O(\varepsilon).$$

Using these two facts, we see that smoothing technique has the same algorithmic complexity, $\widetilde{O}\left( \frac{L D_{\mathcal{Y}}}{\sqrt{\sigma \varepsilon}} \sqrt{\frac{L}{\sigma}} \right)$, as that of DIAG. However the drawback for this method over the direct DIAG is that smoothing technique requires a prefixed tolerance parameter $\varepsilon$.

# E   Experimental details

We consider the following problem.

$$\min_{x \in \mathbb{R}^2} \left[ f(x) = \max_{1 \le i \le m=9} f_i(x) \right] \tag{57}$$

where $f_i(x) = q_{\left(-1,\, (X_i^{(1)}, X_i^{(2)}),\, C_i\right)}(x)$ for all $1 \le i \le 8$, where $q_{(a,b,c)}(x) = a\|x - b\|_2^2 + c$, $X_i^{(1)}$ and $X_i^{(2)}$ are generated from the interval $[-3.0, 3.0]$ uniformly at random, and $C_i$ is generated from the interval $[1.0, 5.0]$ uniformly at random. We fix the last component $f_9(x) = q_{(0.5,\, (0,0),\, 0)}(x)$. Each $f_i$ is smooth with parameter $L = 1$, which implies that $f$ is $L$-weakly convex.

We implement three algorithms: Prox-FDIAG (Algorithm 5), Adaptive Prox-FDIAG (Algorithm 6), and subgradient method [11]. In Prox-FDIAG, we use excessive gap technique [35, Problem (7.11)] (a primal-dual algorithm) to solve the inner sub-problem. As the stopping criteria $\widehat{f}(x_{k+1}; x_k) \le \min_x \widehat{f}(x; x_k) + \tilde{\varepsilon}/4$ cannot be directly checked, we instead check a sufficient condition; we stop the excessive gap technique when the primal-dual gap is less than $\tilde{\varepsilon}/4$, which can be checked efficiently. Adaptive Prox-FDIAG is a variant of Prox-FDIAG, where we adaptively and successively decrease the tolerance parameter $\varepsilon'$ starting from a large tolerance $\varepsilon_0$. It has the same first-order oracle complexity guarantee as Prox-FDIAG (up to an $O(\log(1/\varepsilon))$ factor). However, in Figure 1, we observe that Adaptive Prox-FDIAG can converge faster in practice. We set the initial tolerance $\varepsilon_0$ as $10.0$. For a description of the algorithm we refer to Appendix C.2.

All the algorithms are initialized with the point $x_0 = (4, 4)$ and are given a Lipschitzness parameter of $G = 2L\,\|x_0\|_2$. We run the algorithms ten times with randomly generated instances of the objective function $f(x)$. In Figure 1, we plot the norm of gradient of Moreau envelope $\|\nabla f_{\frac{1}{2L}}(x_k)\|_2$ against the number of iterations $k$ in log-log scale. We compute the gradient of the Moreau envelope at any point $x$, by solving the corresponding convex-concave saddle point problem (18) using Mirror-Prox [34] method with appropriate primal-dual gap based stopping criteria and then using Lemma 4(c). For Prox-FDIAG (red circles), we show in a scatter plot the gradient norm $\|\nabla f_{\frac{1}{2L}}(x_{K(\varepsilon)})\|_2$ at the final output of Prox-FDIAG $x_{K(\varepsilon)}$ versus the total number of inner iterations (of excessive gap technique) taken, for $\varepsilon = 10^0, 10^{-1}, 10^{-2}, 10^{-3}$ over the 10 functions. For Adaptive Prox-FDIAG (black dots) in a scatter plot, we plot the gradient norm $\|\nabla f_{\frac{1}{2L}}(x')\|_2$ at the output $x'$ of each inner sub-problem (excessive gap technique) of each inner Prox-FDIAG step versus the total number of inner iterations (of excessive gap technique) taken to reach that point from the beginning, for $\varepsilon = 10^{-7}$ over the 10 functions. For Prox-FDIAG and Adaptive Prox-FDIAG, using solid red and black (respectively) lines we also plot the best linear function (in log-scale) which fits the scatter points (using default parameters of `scipy.stats.linregress`[2]). For the subgradient method (blue triangles), we plot the mean and standard error of gradient norm $\max_{0 \le k' \le k} \|\nabla f_{\frac{1}{2L}}(x_{\hat{k}(k')})\|_2$ over the 10 instances at iterations $k = 10^0, 10^1, \dots, 10^7$. The estimate at each iteration is the best one so far in the function value, i.e. $\hat{k}(k) \in \arg\min_{0 \le k' \le k} f(x_{k'})$. We see that, Prox-FDIAG and Adaptive Prox-FDIAG have a faster convergence rate than subgradient method. Further, in the same vein as analogous variants in convex non-smooth optimization, Adaptive Prox-FDIAG is faster than Prox-FDIAG almost always.

Subgradient method has a theoretical convergence rate of $O(\frac{1}{\sqrt{K}})$ for a fixed number of iterations $K$ and a constant step-size $\gamma/\sqrt{K+1}$ [11, Corollary 2.2]. However, similar to the case of convex non-smooth problems, we observe that fixed step-size results in a slow convergence. In our experiments, we achieve a faster convergence for the subgradient method by using a diminishing, non-summable but square-summable step-size, $\gamma/\sqrt{k+1}$, which varies with the iteration number $k$. This step-size has convergence rate of $O(\frac{\log(k)}{\sqrt{k}})$ [11, Theorem 2.1], but in practice we observe a faster convergence rate than the constant step-size. After a very simple parameter search, we set $\gamma$ as $0.1 \times G \times L^{3/2}$. We ran subgradient method for a total of $K = 10^7$ number of iterations. Since, subgradient method is not a descent method, at any iteration $k$, we keep track of the best point among all the points we have observed so far, $\{x_0, \cdots, x_{k-1}\}$. Ideally, we should keep track of the point with the minimum norm for the gradient of the Moreau envelope, $\|\nabla f_{\frac{1}{2L}}(x_k)\|_2$, but since the computation of the gradient of Moreau envelope is costly, we only keep track of the point with the minimum function value we have observed so far.

## Footnotes

[2]`https://docs.scipy.org/doc/scipy/reference/generated/scipy.stats.linregress.html`