[Reviews · NeurIPS 2019]

Reviewer 1



This paper aims at solving the saddle point problem $$\min_{x} \max_{y} g(x, y)$$ where $g(x, \cdot)$ is concave for each $x$ and $g(\cdot, y)$ is either strongly convex or nonconvex for every $y$. The authors introduce a new algorithm called DIAG which combines Mirror-prox and Nesterov's AGD to solve the minimax problem. For the case that $g$ is strongly-convex with respect to $x$, the authors show that DIAG has a convergence rate of $\mathcal{O}(1/k^2)$ when the suboptimality for a pair of point $(\hat{x},\hat{y})$ is defined as the primal-dual optimality gap $ \max_{y} g(\hat{x},y) - \min_{x} g(x, \hat{y}) $. For the case that $g$ is nonconvex with respect to $x$, the authors show that DIAG finds an $\epsilon$-first-order stationary point (FOSP) after at most $\mathcal{O}(1/\epsilon^3)$ gradient evaluations. The convergence criteria considered in this paper for solving a strongly convex-concave problem is interesting. In particular, the discussion in the first two paragraphs of Section 3 is very illuminating and clarifies why one cannot achieve a convergence rate of $\mathcal{O}(1/k^2)$ by solving the minimization problem very accurately using any linearly convergent method and using AGD for solving the concave maximization problem. Basically, if one does this procedure the resulted algorithm will achieve a sublinear rate of $O(1/k^2)$ for the error of the concave maximization problem, i.e., $h(y)-h(y^*)$ where $h(y):=\min_{x} g(x, y)$, but this result does not necessarily translate to a convergence rate of $O(1/k^2)$ in terms of primal dual optimality gap. The reviewer believes that this is due to the fact that the function $f(x):=\max_{y} g(x, y)$ may not be smooth in general. I think authors could also add this point to this paragraph. The main idea used for extending the result for the strongly-convex case to the nonconvex setting is to add a proper regularization to the minimization problem to make its objective function strongly convex and use the fact that a first-order stationary point (FOSP) of the regularized problem is close to an FOSP of the original minimization problem. This technique is not novel and has been already used in many papers including reference [27] in the paper. Indeed, by adding this regularization (proximal term) we are in the setting of strongly convex-concave problems and the DIAG method can be used to solve the resulted problem. Overall, the ideas in this paper are interesting (in particular the beginning of Section 3) and the authors did a good job in explaining the challenges for achieving a convergence rate of $\mathcal{O}(1/k^2)$ when $g$ is strongly convex w.r.t. $x$ but concave with respect to $y$. However, the presentation of the algorithm and explanation of the related work are not satisfactory. The reviewer recommends moving the proof for the nonconvex case to the appendix so that the authors will have more space to explain the related work better and present their proposed DIAG method in detail. Below you can find more detailed comments. 1- Section 2.2 of the paper is not well-written. In particular, when the authors talk about the mirror-prox method and the conceptual mirror-prox method (which is also known as Proximal Point method for saddle point problems [R3]) they do not mention that the expressions in (5) and (6) are the special cases of (conceptual) mirror-prox method when we use the Euclidean space. Also, the presentation of sub-routine in (6) for approximately performing (5) is not clear. It would be also helpful to explain that a special case of (6) when one perform two steps of (6) is the Extra-gradient method. 2- In the related work section, the authors mention that the notion of stationarity considered in [26] is weaker than the one used in this paper, and it does not imply the results presented, however no explanation for this statement is provided. 3- The main contribution of the paper is proposing the DIAG method for solving strongly-convex concave saddle point problems. However, the authors do not provide any explanation for DIAG and they just simply provide the Pseudocode of this algorithm. How does DIAG relate to Mirror-Prox? What is the intuition behind the subroutine called ``Imp-STEP"? Why do you need to analyze a weighted average of $x$ while you use the last iterate of $y$ in the analysis? 4- Another concern is the comparison benchmark for the strongly convex-concave case. The authors compare the results to the results of [R2], which only studies the case of convex-concave function. The reviewer feels that this comparison is not fair. Also other papers including [R1] which get a linear convergence rate for the strongly convex-concave case, when the coupling is bilinear have not been cited. 5- Many papers which solve similar problems have not been cited in the paper. For example, papers including [R4] and [R5] (and references therein) should be cited. [R1] Simon S. Du, Wei Hu, ``Linear Convergence of the Primal-Dual Gradient Method for Convex-Concave Saddle Point Problems without Strong Convexit." \textit{AISTATS 2019} [R2] Nemirovski, Arkadi. ``Prox-method with rate of convergence O (1/t) for variational inequalities with Lipschitz continuous monotone operators and smooth convex-concave saddle point problems." \textit{SIAM Journal on Optimization} 15.1 (2004): 229-251. [R3] Aryan Mokhtari, Asuman Ozdaglar, and Sarath Pattathil. ``A unified analysis of extra-gradient and optimistic gradient methods for saddle point problems: Proximal point approach." arXiv preprint arXiv:1901.08511 (2019). [R4] Monteiro, Renato DC, and Benar Fux Svaiter. ``On the complexity of the hybrid proximal extragradient method for the iterates and the ergodic mean." \textit{SIAM Journal on Optimization} 20.6 (2010): 2755-2787. [R5] Maziar Sanjabi, Jimmy Ba, Meisam Razaviyayn, and Jason D Lee. ``On the convergence and robustness of training gans with regularized optimal transport". \textit{In Advances in Neural Information Processing Systems, pages} 7091–7101, 2018. =========== after author's rebuttal ========== I read the comments by the other reviewers as well as the response provided by the authors. The authors did a good job in clarifying the intuition behind their proposed method, and I am fine to recommend acceptance of this paper if the authors include this explanation in the final version. I still think comparing the result in this paper with [[R2] Nemirovski, Arkadi. ``Prox-method with rate of convergence O (1/t) for variational inequalities with Lipschitz continuous monotone operators and smooth convex-concave saddle point problems.] is not fair as [R2] only studies convex-concave problems.

Reviewer 2



This paper studies minimax optimization problem. The target function g(x,y) is assumed to be smooth, i.e., it has Lipschitz gradient wrt (x,y). Two cases are considered: (i) g(x,.) is concave for any fixed x; g(., y) is strongly convex for any fixed y; (ii) g(x,.) is concave for any fixed x; g(.,y) is nonconvex for any fixed y. New algorithms with improved convergence rates are proposed for solving problems in these two classes. In general, I think the results are important. My main concern is that the paper lacks numerical illustration of the performance of the proposed algorithms, and that the authors didn't give any practical examples that satisfy the assumptions of the models. If these can be done, the paper will be much stronger. == after author's rebuttal == My main concerns are well addressed by the authors and I am willing to increase my score.

Reviewer 3



This paper studies smooth saddle-point problems under two different assumptions on the curvature of the problem: (i) the stongly-convex-concave problem and (ii) the nonconvex-concave problem. Regime (i) is rather classical in the field and there are many (also accelerated) algorithms available achieving the optimal rate of convergence in the primal-dual gap function. The second regime is challenging and it is a very interesting and important question to study accelerated methods for such problems. Theorem 1 gives the convergence rate for an accelerated algorithm delivering the optimal convergence rate for the smooth strongly convex-concave saddle point problem. The paper "OPTIMAL PRIMAL-DUAL METHODS FOR A CLASS OF SADDLE POINT PROBLEMS, by YUNMEI CHEN, GUANGHUI LAN, AND YUYUAN OUYANG SIAM J. OPTIM". develops an accelerated primal-dual method for a problem of the form f(x)+\inner{Ax,y}-g(y) where f is smooth convex, and g is convex and non-smooth. Can you relate your result more to this paper and compare your algorithm with theirs? In general I have some hard time understanding the details behind Algorithm 1 (made explicit in the list below) and some other remarks the authors should take care of: -) Line 65: VIs have many more application than just in differential equations. Please provide other references which are more relevant for this audience. -) line 72: Something is missing here when your write liminf S(x). -) The discussion in Section 2.1.2 is a bit misleading. In the end, what you propose here is to search for solving the MinMax problem, so you make a deliberate choice over the MaxMin problem. This is fine, but could be said directly. -) Definition 4 is not complete. It must hold for all $x$ and $y$. -) Lemma 1: Why is it important here that X=R^{p}? -) Eq. (5) no bold letters. -) Instead of referring to the Appendix it might be better to refer to the supplementary materials. -) In Theorem 1 it is not explained what epsilon_step exactly is. Only after the basic complexity bound a specific value for this sequence is provided, but it is still not clear where it appears in the analysis. I guess it is just a tolerance level for executing a subroutine, but this can be explained in the Theorem without having to read the pseudocode. -) I find the pseudocode representation of Algorithm 1 strange. Why are the assignment arrows used here and not simply equalities? -) Algorithm 3 is only defined in the Appendix (supplementary material). It might be good to refer to this in the pseudocode of Algorithm 1. -) I don't understand the subroutine Imp-STEP- What is $R$ exactly and how is it chosen? How is $y_{r}$ computed in eq. (7)? How is x_{k+1} determined from this routine? -) line 196: What do you mean by "the primal-dual gap is unknown". It simply makes no sense to look at it since there is no natural correspondence between a "dual" and a "primal" problem in non-convex optimization. -) line 203: In which sense do you have convergence $O(k^{-1/4})$? -) Lemma 5: First line of the proof should be $\tilde{f}$. -) Proof of Lemma 3: It should read $\tilde{g}(x,y)$.

[Author Response · NeurIPS 2019]

We thank all the reviewers for their constructive comments. We explain the intuition behind DIAG (Algorithm 1) for
strongly-convex-concave minimax problems first, which we will add in the final revision.
**Conceptual DIAG:** The intuition behind Algorithm 1 stems from a "conceptual" version of DIAG (also specified in
Algorithm 1, Step 4), which is inspired from the conceptual version of Mirror-Prox (MP) (cf. Section 2.2):

(a) $w_k = (1 - \tau_k)y_k + \tau_k z_k$

(b) Choose $x_{k+1}, y_{k+1}$ ensuring: $x_{k+1} \in \arg\min_x g(x, y_{k+1})$, and $y_{k+1} = \mathcal{P}_\mathcal{Y}(w_k + \frac{1}{\beta}\nabla_y g(x_{k+1}, w_k))$

(c) $z_{k+1} = \mathcal{P}_\mathcal{Y}(z_k + \eta_k \nabla_y g(x_{k+1}, w_k))$

The main idea is to apply an MP-like update for $x$ on $g(\cdot, y_{k+1})$ and an AGD step for $y$ on $g(x_{k+1}, \cdot)$. In the final
estimate, we use $\bar{x}_K = (2/K(K+1))\sum_{i=1}^K (i\,x_i)$, because MP-like updates give ergodic guarantees, but use $y_K$,
because AGD has final iterate guarantees. The MP-like update is crucial in this algorithm so as to inherit the well-known
fast convergence rate of AGD for smooth-convex optimization.
**Implementable DIAG:** The above step (b) requires $g(\cdot, y_{k+1})$ and $g(x_{k+1}, \cdot)$ which are not a priori available at the $k$-th
step. But we can implement this step up to $\varepsilon_{\text{step}}$ error (step 4, Algorithm 1), using `Imp-STEP` subroutine (Algorithm
1). Just like the fact that conceptual MP can be realized in $\log(1/\varepsilon)$ steps (in fact, just two steps suffice), `Imp-STEP`
converges in $R = \log(\frac{2D_\mathcal{Y}}{\varepsilon_{\text{mp}}}) = O(\log(\frac{1}{\varepsilon_{\text{step}}}))$ steps, because the following mapping is a contraction for small enough
stepsize $1/\beta$:

$$y^{i+1} = \mathcal{P}_\mathcal{Y}(w_k + (1/\beta)\nabla_y g(x^*(y^i), w_k)),\tag{1}$$

where $x^*(y) = \arg\min_x g(x, y)$. This follows from (i) the $L$-smoothness of $g$, and (ii) the Lipschitzness of $x^*(y)$ in $y$
(due to strong convexity of $g(\cdot, y)$). Further, again by $\sigma$-strong-convexity of $g(\cdot, y)$, $x^*(y) = \arg\min_x g(x, y)$ could be
approximately found in $O(\sqrt{\frac{L}{\sigma}}\log(\frac{1}{\varepsilon_{\text{step}}}))$ steps. Thus the overall speed of `Imp-STEP` is $O(\sqrt{\frac{L}{\sigma}}\log^2(\frac{1}{\varepsilon_{\text{step}}}))$ steps.
**Response to reviewer 1:** We agree with and will include, the reviewer's comment, that the non-smoothness of
$f(x) = \max_y g(x, y)$, more precisely the non-Lipschitzness of the maximizer of $g(x, \cdot)$ is the reason why naive AGD
is sub-optimal. We will devote more space to explaining the DIAG algorithm and discussing more related works.
1- We will clarify that steps (5) & (6) is the Euclidean version of Mirror-Prox and discuss the extra-gradient method.
2- Criteria in [26] is weaker in the following sense. Consider $g(x, y) = (x^2 - y^2)/2$ ( $f(x) = x^2/2$, $h(y) = -y^2/2$ )
with domain $\mathbb{R} \times [0, 1]$. To reach $(\hat{x}, \hat{y})$ s.t. $\hat{x} = \hat{y} \le \epsilon$, DIAG requires $O(\varepsilon^{-3})$ steps since $\nabla f(\hat{x}) = \nabla h(\hat{y}) = \varepsilon$, how-
ever, [26] requires $O((\varepsilon^2)^{-3.5}) = O(\varepsilon^{-7})$ steps since $\mathcal{Y}(\hat{x}, \hat{y}) = \max_{y' \in [0,1]} \langle \nabla_y g(\hat{x}, \hat{y}), y' - \hat{y} \rangle = \langle -\varepsilon, -\varepsilon \rangle = \varepsilon^2$.
We will add a precise justification (which was omitted due to the lack of space) in the next revision.
3- We refer the reviewer to the above explanation of DIAG algorithm.
4- **Bilinear** coupling: a) we focus on non-linear coupling and in general, bilinear results do not apply to our setting, b)
when we specialize our result to standard bilinear coupling setting, our results match the optimal $1/K^2$ rates. Further
assumptions like unbounded domain and full-rank coupling matrix give linear convergence rates [R1] (will be cited),
but this follows directly from the fact that the Fenchel dual of a smooth function is strongly convex (Theorem 6 of [12]).
5- We will include citations to similar saddle point problems and algorithms, including [R4] and [R5]. However,
we again note that none of the suggested (or other) references obtain results similar to ours in the setting that we consider.
35
**Response to reviewer 3:** We will include numerical experiments; as a preliminary
experiment we consider the following min-max problem (P3): $\min_{x \in \mathbb{R}^2} \left[ f(x) = \right.$
$\left. \max_{1 \le i \le m=9} f_i(x) \right]$ with random quadratic functions (hence weakly-convex). In the
figure right, we plot the norm of gradient of Moreau envelope $\|\nabla f_{\frac{1}{2L}}(x_k)\|_2$ against the
number of first-order gradient oracle calls in log-log scale. We see that, Prox-FDIAG has
a faster convergence rate than subgradient method. We will also include other practical
use-cases such as robust learning, multi-task learning, and adversarial training.

**Response to reviewer 4:** We will incorporate all suggestions by the reviewer and clarify all ambiguous/missing
explanations in the final version. We discuss important ones below.
-*Chen et al.*: their result only handles bilinear case (also see response to R1, point 4) and gets a rate of $O(1/\epsilon)$, but can
handle prox-function friendly non-smoothness w.r.t. $y$. In contrast, we can handle non-linear coupling between $x, y$ and
for bilinear case (with strong convexity w.r.t. $x$ and smoothness w.r.t. $y$) can obtain $O(1/\sqrt{\epsilon})$ rate.
-) We assume $\mathcal{X} = \mathbb{R}^p$ since we use [Theorem 6, 12] in the proof, which requires the domain to be the full vector space.
-) The sub-routine Imp-STEP has a typo: In Step 10, $x_r$ should be $\hat{x}_r$. That is, given $y_r$ we compute $\hat{x}_r$ such that
$g(\hat{x}_r, y_r) \le \min_x g(x, y_r) + \varepsilon_{\text{agd}}$ and then Step 11 updates: $y_{r+1} = \mathcal{P}_\mathcal{Y}(w + \frac{1}{\beta}\nabla_y g(\hat{x}_r, w))$. This gives the new
$(\hat{x}_r, y_{r+1})$ pair, and the process is repeated. We refer the reviewer to the explanation of DIAG algorithm at the top.
-) In line 196: We meant that $\min_x \max_y g(x, y) - \max_y \min_x g(x, y)$ (which we call the minimum primal dual gap)
is unknown for non-convex functions. We will make the statement precise.
-) In line 203: We are citing the result of [8], which uses the same convergence criteria as our paper.

[Meta-Review · NeurIPS 2019]

All reviewers agreed that this paper makes an interesting contribution to NeurIPS. Please make sure to take the reviewers' comments in consideration for the camera-ready version, in particular improving the clarity of the presentation and making the overall statements more precise.